# Robust Decision Aggregation with Second-order Information

Submission Id: 419

## ABSTRACT

We consider a decision aggregation problem with two experts who each make a binary recommendation after observing a private signal about an unknown binary world state. An agent, who does not know the joint information structure between signals and states, sees the experts' recommendations and aims to match the action with the true state. Under the scenario, we study whether supplemented additionally with second-order information (each expert's forecast on the other's recommendation) could enable a better aggregation.

We adopt a minimax regret framework to evaluate the aggregator's performance, by comparing it to an omniscient benchmark that knows the joint information structure. With general information structures, we show that second-order information provides no benefit – no aggregator can improve over a trivial aggregator, which always follows the first expert's recommendation. However, positive results emerge when we assume experts' signals are conditionally independent given the world state. When the aggregator is deterministic, we present a robust aggregator that leverages second-order information, which can significantly outperform counterparts without it. Second, when two experts are homogeneous, by adding a non-degenerate assumption on the signals, we demonstrate that random aggregators using second-order information can surpass optimal ones without it. In the remaining settings, the second-order information is not beneficial. We also extend the above results to the setting when the aggregator's utility function is more general.

## CCS CONCEPTS

• **Theory of computation → Algorithmic mechanism design**; *Algorithmic game theory*.

## KEYWORDS

Decision Aggregation, Second-order Information, Robust Aggregation

**ACM Reference Format:**
Anonymous Author(s). 2023. Robust Decision Aggregation with Second-order Information. In *Proceedings of Make sure to enter the correct conference title from your rights confirmation email (Conference acronym 'XX)*. ACM, New York, NY, USA, 27 pages. https://doi.org/XXXXXXX.XXXXXXX

## 1 INTRODUCTION

Two marketing experts at a company are providing advice on whether to launch a new product now or delay the launch to next

year. The unknown binary world state is whether the market has a high demand for the product now or a low demand currently. Each expert can recommend "launch now" or "delay launch" as the optimal action. If the action matches the true demand (launch now when high demand, or delay when low demand), the utility of the company is +1. If mismatched, the utility is −1.

The agent takes the experts' recommendations into consideration and outputs an aggregated decision. If both experts recommend the same action, the agent can follow this consensus recommendation. When they suggest opposite actions, the aggregator may follow the expert who has superior past accuracy. However, in the one-shot setting, the aggregator would fall into a dilemma without access to performance history. Like the above motivating example, similar dilemmas universally exist in other scenarios, e.g., when a patient faces different diagnoses from two doctors, when an investor faces diverse opinions on a startup company, and when an editor faces conflicting recommendations from two referees.

Now, let us ask each expert to provide a prediction about their peer's recommendation, called the second-order setting. Back to our motivating example, for instance, the first expert recommends "launch now" and predicts her peer recommends "launch now" with a probability of 0.4, and the second expert recommends "delay launch" and predicts her peer recommends "launch now" with a probability of 0.2. Our question is – *by having each expert additionally provide a prediction about their peer's recommendation, can we better aggregate their recommendations?*

A series of works have demonstrated the benefit of the additional second-order information in the information aggregation problem. The most closed setting is considered by Prelec et al. [24], proving that second-order information enhances aggregation given a sufficiently large group size. However, the value of second-order information remains less explored for smaller expert groups.

In particular, prior analyses considered settings where second-order information identifies the true world state, even absent prior data. However, with only two experts, the information often cannot unambiguously determine the state. In such a case, we adopt a robust aggregation paradigm to evaluate the aggregator performance against an omniscient benchmark. The paradigm is introduced by Arieli et al. [1]. The omniscient agent observes the experts' private signals and knows the true joint distribution between signals and states. This allows a perfect aggregation to output the optimal recommendation. In contrast, we focus on aggregators who only know the family of possible information structures, not the exact structure itself. The goal is to identify an aggregation rule with minimum regret compared to the omniscient benchmark. Here, regret is defined as the worst-case difference between the benchmark's expected utility and the aggregator's expected utility among all information structures.

Under the robust aggregation paradigm, we will identify the optimal aggregator that only uses the experts' first-order recommendations; and the optimal aggregator that uses both the first-order

recommendations and second-order predictions. By comparing the regret achieved by the optimal aggregators with and without second-order predictions, we can quantify the value of the additional higher-order information for robust aggregation.

## 1.1 Summary of Results

As indicated by the motivating example, our primary focus lies in the scenario where two experts are asked to provide recommendations for binary actions after observing a binary signal, with the agent aiming to align the action with a binary world state.

*General information structures.* We initially make no assumption on the underlying information structure, allowing for the possibility of a strong conditional correlation between the signals observed by the two experts. In this context, we present a negative result, demonstrating that no random aggregator can ensure a regret below 0.5, even when equipped with second-order information (Theorem 3.1). Further, this regret can be guaranteed by a trivial aggregator that consistently follows the first expert's recommendation (Theorem 3.3). Consequently, the second-order information does not provide any assistance in constructing a robust aggregator under these circumstances.

*Conditionally independent information structures.* Subsequently, we narrow our focus to conditionally independent information structures to reveal the power of second-order information. Here, conditionally independent information structures mean that two experts' signals are independent conditioning on the world state. We start with no additional assumptions, allowing for heterogeneous experts. We then consider homogeneous experts, meaning that two experts have an identical marginal signal distribution. Building on this, we further assume non-degenerate signals, i.e., experts recommend different actions after seeing distinct signals. On the aggregator side, we consider two kinds: (1) deterministic aggregators that output a fixed action, and (2) random aggregators that output a random action according to a probability distribution over actions.

Two key positive results emerge. First, for deterministic aggregators, significant improvements occur in the general conditionally independent setting. Second, for random aggregators, we demonstrate that second-order information enables lower regret guarantees under the assumptions of homogeneous experts with non-degenerate signals. The remainder of the results are negative, i.e., second-order information cannot enhance the robustness of the aggregator. All results are presented in Table 1. We now discuss these results in more detail.

*1. Heterogeneous experts.* We first consider the general scenario where two experts can be heterogeneous. In this context, with respect to deterministic aggregators, we present a positive result. We construct an aggregator that adheres to the more "informative" expert when two experts' recommendations split. For a better understanding, the more "informative" expert has better accuracy in predicting the other's recommendation[1]. We show that such an aggregator guarantees a regret of $1/3 \approx 0.3333$, which is

---

[1]Here accuracy is measured by the distance between the prediction and recommendation. For instance, predicting 0.7 is more accurate than predicting 0.6 for actual recommended action 1.

the most robust deterministic aggregator with second-order information (Theorems 4.3 and 4.4). Furthermore, it is essential to note that no aggregator can guarantee a regret lower than 0.5 without second-order information. This highlights the substantial assistance offered by second-order information (Theorem 4.1).

However, considering random strategies, second-order information proves to be redundant in terms of robustness. In particular, no aggregator equipped with second-order information can ensure a regret of less than 0.25 (Theorem 4.5); while such a regret can also be achieved by a simple aggregator that uniformly chooses an action in cases of different recommendations (Theorem 4.6).

To summarize, when facing heterogeneous experts, under the robust aggregation paradigm, (1) when the aggregator is deterministic, second-order information can significantly enhance its decision-making capabilities; (2) when the aggregator can be random, second-order information is not beneficial to the regret.

*2. Homogeneous experts.* We also investigate the scenarios where two experts are homogeneous. We show that without further assumption, neither deterministic aggregators (Theorems 5.2 and 5.4) nor random aggregators (Theorems 5.5 and 5.6) can derive benefits from second-order information. This is due to a special case where both experts always recommend the same action, rendering the second-order information useless. We show that such a case is the worst one, and could lead to a tight regret lower bound of $3 - 2\sqrt{2} \approx 0.1716$.

*3. Homogeneous experts with non-degenerate signals.* To avoid the above special case, we focus on the information structures where experts will recommend different actions when observing different signals. In this setting, we encounter a negative outcome considering deterministic strategies: no aggregator with second-order information can surpass the performance of the follow-the-first-expert aggregator, which guarantees a regret of $3 - 2\sqrt{2}$ (Theorem 5.4 and Corollary 6.2). Notably, when facing homogeneous experts, this aggregator is equivalent to the uniform aggregator, which uniformly chooses an action when two recommendations conflict.

However, when considering random aggregators, we provide two aggregators leveraging second-order information that give better performance. The first aggregator follows a similar principle to the robust deterministic aggregator when dealing with heterogeneous experts, i.e., granting more weight to the expert with more accurate predictions. This aggregator guarantees a regret of 0.1682 (Theorem 6.10). The second aggregator, derived from the online learning algorithm proposed by Guo et al. [11], guarantees an even lower regret of 0.1673 (Theorem 6.11). Both of these aggregators outperform the uniform aggregator with a regret of $3 - 2\sqrt{2} \approx 0.1716$, which is already optimal without second-order information (Theorem 6.5). This positive result aligns with the findings in Prelec et al. [24], which highlights that second-order information can enable the aggregator to achieve no regret when confronted with infinite experts. In our investigation, we extend this positive outcome to the case of two experts. We also establish a lower bound of $1/6 \approx 0.1667$ for aggregators with second-order information (Theorem 6.7).

*Extension: general utility functions with homogeneous experts.* In Section 7 and Appendix A, we also extend the above findings for homogeneous experts to encompass general utility functions, where the agent's objective goes beyond aligning actions with

**Table 1: An overview of our main results.**

| Regret lower/upper bound[*] | | Deterministic | Helps? | Random | Helps? |
|---|---|---|---|---|---|
| Heterogeneous | 1st[†] | 0.5 | Yes | 0.25 | No |
| | 2nd[†] | 0.3333[‡] | | 0.25 | |
| Homogeneous | 1st | 0.1716[‡] | No | 0.1716 | No |
| | 2nd | 0.1716 | | 0.1716 | |
| Homogeneous & Non-degenerate | 1st | 0.1716 | No | 0.1716 | Yes |
| | 2nd | 0.1716 | | [0.1667, 0.1673][‡] | |

[*] When the entry is a single value, it means the lower/upper bound coincides at the value; when the entry is an interval, the endpoints of the interval respectively represent the lower/upper bound.

[†] 1st: only using first-order recommendations, 2nd: also using second-order predictions.

[‡] $0.3333 = 1/3$, $0.1716 = 3 - 2\sqrt{2}$, $0.1667 = 1/6$, $0.1673$ is a numerically rounded estimate.

states. We first perform a reduction to allow us to focus solely on the ratio of the utility gap between adopting two actions when the state is 0 and when the state is 1. We observe that the results for random aggregators mirror those in the previous setting. Without further assumptions, the previous negative outcome still holds when both experts always advocate an identical action, resulting in high regret for all aggregators. However, when we introduce the non-degenerate signal assumption, we demonstrate that second-order information enhances aggregators' robustness with different ratios. These findings underscore the significance of second-order information for a wide range of utility functions.

## 1.2 Related Work

Our work focuses on decision aggregation, a subset of the broader field of information aggregation. A significant portion of information aggregation literature focuses on forecast aggregation. This body of work explores various methodologies, including simple techniques like averaging [8], median averaging [14], and their respective modifications [5, 17, 25]. These studies showcase the efficacy of straightforward aggregation rules, such as averaging or random dictating [3, 8, 10, 20, 28], mirroring some of our findings. Furthermore, there exists a body of literature on decision aggregation, such as De Oliveira et al. [9], Arieli et al. [3], and Prelec et al. [24], which closely align with our work.

The most closely related paper is Prelec et al. [24], as it also examines the role of second-order information in decision aggregation. The key distinction from our setting is their focus on infinite, homogeneous experts. Leveraging second-order information, they develop an aggregator that identifies the true world state, i.e., has no regret compared to the omniscient agent. In contrast, we focus on two heterogeneous experts, representing a small expert group. We demonstrate that for such settings, regret is unavoidable even with second-order information. Further, we characterize cases where second-order information does not help reduce regret. Our regret analysis and findings on the limitations of small heterogeneous groups add new insights into decision aggregation with second-order information.

We adopt a regret-based minimax paradigm in our analysis. While De Oliveira et al. [9] also employ a minimax approach, they use a loss-based framework and show the optimal aggregator simply follows the best expert. In contrast, we evaluate aggregator performance in comparison to an omniscient benchmark using a regret formulation. This regret-based robust paradigm follows the approach of Arieli et al. [1], which studied forecast aggregation under Blackwell-ordered and conditionally independent structures. Additional works employing the regret-based approach include Babichenko and Garber [4], which explores partial-evidence information structures within a repeated game context, and Guo et al. [11], which proposes an algorithmic framework for robust forecast aggregation.

Our model focuses on the setting where the aggregator does not have access to the exact information structure. Many other works also consider settings with ambiguity but differ in the knowledge they assume the aggregator possesses. For instance, both De Oliveira et al. [9] and Arieli et al. [3] assume that the aggregator possesses knowledge of the marginal distribution of each expert's signal and aim to enhance the robustness of correlations based on this information. This consideration is also prevalent in [6, 12, 18]. In contrast, our work assumes that the aggregator is ignorant about the full information structure and solely has access to the experts' outputs and the set to which the information structure belongs. This approach aligns with Arieli et al. [1], Kong [15], and Prelec et al. [24]. Moreover, Arieli et al. [2] characterize the set of identifiable information structures and introduce a scheme that uniquely identifies the state of nature in finite cases.

We study whether second-order information improves the performance of the decision aggregator. A substantial body of literature also explores the benefit of second-order information. Prelec [23] started the exploration and introduced a framework wherein agents provide both their answers and predictions for a single multi-choice question. Building upon this foundation, subsequent works have delved into the design of aggregators utilizing second-order information. Among these, the "surprisingly popular" approach, initially introduced by Prelec et al. [24] and subsequently developed by researchers such as Palley and Soll [22], Palley and Satopää [21], and Chen et al. [7], have garnered significant attention.

Furthermore, Kong [15] employs second-order and even higher-order information in forecast aggregation, particularly in scenarios featuring two experts who are either Blackwell-ordered or receive signals that are conditionally independent and identically distributed. Wang et al. [29], Martinie et al. [19], and Wilkening et al. [30] have designed prediction-aided forecast aggregators and

conducted experimental analyses to showcase their effectiveness and potential in real-world applications.

In addition to decision aggregation and forecast aggregation, the utilization of second-order information finds application in many other domains. For instance, Kong et al. [16] focus on open-response questions and ask agents what they think other people will answer. They use the information to rank the answers without any prior knowledge. Also, Hosseini et al. [13] and Schoenebeck and Tao [27] employ a similar framework to rank a predefined set of candidates. In the context of election forecasting, Rothschild and Wolfers [26] utilize voters' expectations regarding other people's votes to provide more accurate predictions of election outcomes.

## 2 PROBLEM STATEMENT

A company is deciding whether to launch a new product now (action 1) or delay it until next year (action 0). There are two marketing experts, expert 1 and expert 2, providing recommendations to the company's CEO (the agent). There are two possible world states about the current demand, high (state 1) and low (state 0), and we use $\omega \in \Omega = \{0, 1\}$ to denote the unknown true state. Let $a \in A = \{0, 1\}$ denote the action adopted by the CEO. If the action matches the true demand (launch now with high demand or delay with low demand), the CEO's utility is $+1$. Otherwise, the CEO's utility is $-1$.

Each expert $i \in \{1, 2\}$ receives a private signal $S_i \in S = \{L, H\}$ about the demand, where an $L$ signal implies a lower likelihood of the high demand state than an $H$ signal. The realization of $S_i$ is $s_i$. An information structure $\pi \in \Delta(\Omega \times S^2)$ is a joint distribution of the state and two private signals $S_1, S_2$, which encodes the correlation between the true state and private signals. Here, $\Delta(\cdot)$ stands for the set of all distributions on the support. The information structure is shared by both experts. Thus, the prior of the world state $\mu = \pi(\omega = 1) \in [0, 1]$ is also known to both experts. For simplicity, we rewrite the key parameters of any specific information structure in the rest of this paper. Let

$$k_1 := \pi(S_1 = L \mid \omega = 1), \quad l_1 := \pi(S_1 = L \mid \omega = 0);$$
$$k_2 := \pi(S_2 = L \mid \omega = 1), \quad l_2 := \pi(S_2 = L \mid \omega = 0).$$

denote the signal probabilities and the posteriors are written as

$$b_{1L} := \pi(\omega = 1 \mid S_1 = L) \le b_{1H} := \pi(\omega = 1 \mid S_1 = H);$$
$$b_{2L} := \pi(\omega = 1 \mid S_2 = L) \le b_{2H} := \pi(\omega = 1 \mid S_2 = H).$$

Here, $\pi(S_1 = L \mid \omega = 1)$ stands for the probability that $S_1 = L$ conditioning on $\omega = 1$. Similar meanings hold for similar notations. Note that $b_{1L} \le b_{1H}$ and $b_{2L} \le b_{2H}$ hold since an $L$ signal indicates a lower likelihood of the high demand state 1.

The experts observe their private signals and compute posteriors on the demand state. They recommend "launch now" (action 1) if the posterior on the high demand $\ge 0.5$ and "delay launch" (action 0) otherwise. We assume that experts report truthfully. Such incentive compatibility can be guaranteed by rewarding the experts later with the revelation of the true state.

The CEO aims to aggregate the expert recommendations into an optimal product launch decision but lacks knowledge of the underlying information structure. Formally, the CEO observes the recommendations $a_1, a_2 \in A = \{0, 1\}$ from the two marketing experts. The CEO's aggregator outputs a final decision, which may be deterministic (denoted by $f^d : A^2 \to A$) or randomized (denoted by $f^r : A^2 \to \Delta(A)$). Here $\Delta$ means its output is a random action following a probability distribution.

*Benchmark and regret.* We compare the aggregator to an omniscient agent who knows the true information structure $\pi$ and observes the realized signals $s_1, s_2$. The benchmark's output is

$$a^*(s_1, s_2, \pi) := \phi(\pi(\omega = 1 \mid S_1 = s_1, S_2 = s_2)),$$

where $\phi(b) := \mathbb{1}\{b \ge 0.5\}$. $\mathbb{1}\{\cdot\}$ is the indicator which is valued 1 when the inner condition is true and 0 otherwise.

The loss of any aggregator regarding information structure $\pi$ is defined by the benchmark's expected utility subtracted by the aggregator's expected utility:

$$L(f, \pi) := \mathbb{E}\left[\mathbb{1}\{a^*(s_1, s_2, \pi) = \omega\} - \mathbb{1}\{f(a_1(s_1), a_2(s_2)) = \omega\}\right].$$

Here, the expectation is taken on the world state $\omega$, experts' private signal realizations $s_1, s_2$, and the randomness of the aggregator's output, if it is random.

*Second-order information.* Each expert, denoted as $i$, is also asked to provide a prediction, $p_i \in [0, 1]$, for the probability that the other expert recommends action 1. This is represented as follows:

$$p_1 := \mathbb{E}_{s_1}[\phi(b_2) = 1 \mid S_1 = s_1]$$
$$= \sum_{s_2} \pi(S_2 = s_2 \mid S_1 = s_1)\phi(\pi(\omega = 1 \mid S_2 = s_2)),$$

and $p_2$ is defined analogously. The CEO aims to find an aggregator that can effectively incorporate this additional second-order information. Such an aggregator is denoted as $f : A^2 \times [0, 1]^2 \to \Delta(A)$ or $A$. We distinguish between deterministic aggregators $f^d$ with deterministic outputs, and randomized aggregators $f^r$ that output probability distributions over actions.

Given the information structure $\pi$, the loss of any aggregator equipped with second-order information is defined as follows:

$$L(f, \pi) := \mathbb{E}\left[\mathbb{1}\{a^*(s_1, s_2, \pi) = \omega\} - \mathbb{1}\{f(a_1, a_2, p_1, p_2) = \omega\}\right].$$

Here, $a_1, a_2, p_1, p_2$ are abbreviations for $a_1(s_1), a_2(s_2), p_1(s_1), p_2(s_1)$, respectively. Again, the expectation is taken on the state $\omega$, experts' private signal realizations $s_1, s_2$, and the randomness of the aggregator's output, if it is random.

*Robust aggregation.* As the CEO lacks knowledge of the exact information structure, her goal is to design an aggregator that performs well across all possible information structures within the family $P$. To model such robustness, we define the *regret* of a deterministic aggregator $f$ as the worst-case loss across all information structures in $P$:

$$L_P(f) := \max_{\pi \in P} L(f, \pi).$$

This work considers different sets of information structures to capture various real-world scenarios. For each of these information structure family $P$, our objective is to identify the best aggregators $f$, both with and without second-order information, that minimize the regret $L_P(f)$. We denote the set of all deterministic aggregators without second-order information as $F_{+1}$, and the set of deterministic aggregators with second-order information as $F_{+2}$. Formally,

we address the following optimization problems:

$$\min_{f^d \in F_{+1}} L_P(f^d) = \min_{f^d \in F_{+1}} \max_{\pi \in P} L(f^d, \pi),$$

$$\min_{f^d \in F_{+2}} L_P(f^d) = \min_{f^d \in F_{+2}} \max_{\pi \in P} L(f^d, \pi),$$

$$\min_{f^r \in \Delta(F_{+1})} L_P(f^r) = \min_{f^r \in \Delta(F_{+1})} \max_{\pi \in P} L(f^r, \pi),$$

$$\min_{f^r \in \Delta(F_{+2})} L_P(f^r) = \min_{f^r \in \Delta(F_{+2})} \max_{\pi \in P} L(f^r, \pi).$$

We further study whether the inclusion of second-order information leads to a strict decrease in regret for the agent. In other words, we examine for different family $P$ whether the following two values $< 0$:

$$\min_{f^d \in F_{+2}} L_P(f^d) - \min_{f^d \in F_{+1}} L_P(f^d),$$

$$\min_{f^r \in \Delta(F_{+2})} L_P(f^r) - \min_{f^r \in \Delta(F_{+1})} L_P(f^r).$$

## 3 WARM-UP: GENERAL INFORMATION STRUCTURES

As a warm-up, this section examines the information structure family ALL, which contains all information structures that encode the correlation between the state and two signals. Missing proofs of this section can be found in Appendix B. We start by presenting a universal lower bound.

**Theorem 3.1.** *For every random aggregator $f^r(a_1, a_2, p_1, p_2) \in \Delta(F_{+2})$, $L_{ALL}(f^r) \geq 0.5$.*

To prove the above lower bound, we adapt Yao's principle [31] to our setting, establishing a connection between the expected regret of any random aggregator and the best aggregator for any distribution over information structures. The lemma below will be invoked repeatedly in the subsequent sections.

**Lemma 3.2** (Yao's principle [31]). *In any aggregator family $F$ and information structure family $P$, for any random aggregator $f^r \in \Delta(F)$ and any distribution $D \in \Delta(P)$,*

$$\min_{f^d \in F} \mathbb{E}_{\pi \sim D}[L(f^d, \pi)] \leq \max_{\pi \in P} \mathbb{E}_{f^d \sim f^r}[L(f^d, \pi)].$$

We now present a deterministic aggregator in $F_{+1}$ that achieves for ALL the lowest regret among all aggregators in $\Delta(F_{+2})$, which we refer to as the follow-the-first-expert aggregator.

*The follow-the-first-expert aggregator.* The aggregator is characterized by unconditionally adhering to the recommended action of expert 1, irrespective of expert 2's advice. This aggregator can be mathematically expressed as $f_{ftfe}(a_1, a_2) = a_1$. We have the following regret guarantee for $f_{ftfe}$.

**Theorem 3.3.** $L_{ALL}(f_{ftfe}) = 0.5$.

Since $f_{ftfe}$ is optimal among all aggregators in $\Delta(F_{+2})$ and is itself in $F_{+1}$, we conclude that the agent cannot reach a lower regret when two experts have conditional correlations even by using a random strategy or incorporating second-order information. We therefore turn our focus to conditionally independent information structures in the following sections.

## 4 HETEROGENEOUS EXPERTS

We now come to consider conditionally independent information structures and make no additional assumptions on experts, allowing them to be heterogeneous. Specifically, a conditionally independent information structure ensures that two experts' signals are independent given the state. The set of all conditionally independent information structures is referred to as CI. Missing proofs of this section can be found in Appendix C.

### 4.1 Deterministic Aggregators

We first establish the following lower bound for deterministic aggregators.

**Theorem 4.1.** *For every deterministic aggregator $f^d(a_1, a_2) \in F_{+1}$, $L_{CI}(f^d) \geq 0.5$.*

We now revisit the follow-the-first-expert aggregator introduced in Section 3. Since $CI \subset ALL$, as a corollary of Theorems 3.3 and 4.1, within CI, this aggregator is still the optimal among all deterministic aggregators.

**Corollary 4.2.** $L_{CI}(f_{ftfe}) = 0.5$.

We proceed to establish a lower bound for deterministic aggregators equipped with second-order information, which notably falls far below the lower bound for deterministic aggregators lacking second-order information.

**Theorem 4.3.** *For every deterministic aggregator $f^d(a_1, a_2, p_1, p_2) \in F_{+2}$, $L_{CI}(f^d) \geq 1/3 \approx 0.3333$.*

To further establish the effect of second-order information in this setting, we now introduce a robust "threshold aggregator". Remarkably, within CI, $f_{thr}$ attains the lowest regret among all deterministic aggregators in $F_{+2}$. This observation underscores the potential of prediction knowledge in enabling an agent without randomness to achieve a lower regret in CI.

*The threshold aggregator.* This aggregator follows experts' recommendations if the experts agree. When the experts disagree, it compares the sum of their predictions to 1. If the predictions sum to less than 1, it chooses action 1. Otherwise, it chooses action 0. Concretely, we have

$$f_{thr}(a_1, a_2, p_1, p_2) = \begin{cases} a_1 & a_1 = a_2 \\ 1 & a_1 \neq a_2, p_1 + p_2 \leq 1 \\ 0 & a_1 \neq a_2, p_1 + p_2 > 1 \end{cases}.$$

The aggregator tends to trust the expert who makes a more accurate prediction. From another perspective, it is equivalent to the "surprisingly popular" approach proposed by Prelec et al. [24]. When $p_1 + p_2 \leq 1$, the answer 1 is more popular than predicted. Conversely, when $p_1 + p_2 > 1$, the answer 0 is the surprisingly popular one. We demonstrate that the threshold aggregator has a regret of 1/3.

**Theorem 4.4.** $L_{CI}(f_{thr}) = 1/3$.

In summary, the threshold aggregator resolves the experts' disagreement based on who can predict the other more accurately.

While there still exists a gap compared with the omniscient benchmark, using second-order information already significantly improves the aggregator's performance versus relying solely on raw recommendations.

## 4.2 Random Aggregators

For random aggregators, we first establish that it is impossible to ensure a regret below 0.25 even when utilizing second-order information.

**Theorem 4.5.** *For every random aggregator $f^r(a_1, a_2, p_1, p_2) \in \Delta(F_{+2})$, $L_{CI}(f^r) \geq 0.25$.*

We now introduce a random aggregator that does not require predictive information yet still guarantees tight regret. We refer to the aggregator as the uniform aggregator. It can be seen as a random version of the follow-the-first-expert aggregator.

*The uniform aggregator.* The uniform aggregator outputs the recommendations of experts when they agree; when they disagree, the aggregator uniformly selects an action. In other words, the uniform aggregator can be expressed as

$$f_{uni}(a_1, a_2) = \begin{cases} a_1 & a_1 = a_2 \\ 0.5 & a_1 \neq a_2 \end{cases}.$$

Here, when $f_{uni}$ outputs 0.5, it means choosing action 1 with probability 0.5. A similar interpretation also holds for random aggregators to be introduced later.

**Theorem 4.6.** $L_{CI}(f_{uni}) = 0.25$.

## 5 HOMOGENEOUS EXPERTS

In the above results for random aggregators, an important reason why the predictions are useless is that, in the worst cases, they do not assist the agent in distinguishing the omniscient expert from the ignorant one, thus the aggregator can only choose the uniform strategy at best. However, the benchmark, aided by the information structure, is always able to identify the more informed expert. As a result, there exists a substantial utility gap between the agent and the benchmark, irrespective of the agent possessing knowledge of the prediction.

In this section, we assume that the two experts are homogeneous, which means their marginal signal distribution is the same. Intuitively, the knowledge of prediction may help the agent identify the representative signal that includes the information of the real state. Formally, we take the following assumption in this section.

**Assumption 5.1** (Homogeneous experts). *The two experts are homogeneous. In other words, $k_1 = k_2 = k$, $l_1 = l_2 = l$, $b_{1L} = b_{2L} = b_L$ and $b_{1H} = b_{2H} = b_H$.*

We then focus on the set of all conditionally independent information structures with homogeneous experts, which is referred to as HOI. All missing proofs of this section are deferred to Appendix D.

## 5.1 Deterministic Aggregators

To begin, we establish a lower bound for deterministic aggregators, demonstrating that no deterministic aggregator in $F_{+2}$ can achieve a regret less than $3 - 2\sqrt{2} \approx 0.1716$.

**Theorem 5.2.** *For every deterministic aggregator $f^d(a_1, a_2, p_1, p_2) \in F_{+2}$, $L_{HOI}(f^d) \geq 3 - 2\sqrt{2}$.*

Interestingly, the threshold aggregator, which is optimal with heterogeneous experts, achieves suboptimal performance in the homogeneous setting, still giving a regret of $1/3$.

**Theorem 5.3.** $L_{HOI}(f_{thr}) = 1/3$.

Nevertheless, the follow-the-first-expert aggregator can guarantee a lower regret. Moreover, $f_{ftfe}$ achieves the lowest regret among all deterministic aggregators in $F_{+2}$ regarding HOI. This also indicates that knowledge of prediction cannot help the agent without randomness attain a lower regret in HOI.

**Theorem 5.4.** $L_{HOI}(f_{ftfe}) = 3 - 2\sqrt{2}$.

## 5.2 Random Aggregators

For random aggregators utilizing second-order information, we first show that their regret lower bound in HOI is $3 - 2\sqrt{2}$.

**Theorem 5.5.** *For every random aggregator $f^r(a_1, a_2, p_1, p_2) \in \Delta(F_{+2})$, $L_{HOI}(f^r) \geq 3 - 2\sqrt{2}$.*

As an intuition of the proof, we provide an information structure in which the aggregator always observes the input $(1, 1, 1, 1)$, which means the best strategy is to adopt action 1 all the time. However, when two signals are both $L$, the benchmark's posterior is less than 1. Therefore, no aggregator can avoid such a difference, leading to an unavoidable regret of $3 - 2\sqrt{2}$.

We then show that the uniform aggregator is optimal with a tight regret of $3-2\sqrt{2}$. We notice here that since experts are homogeneous, the uniform aggregator is equivalent to the follow-the-first-expert-aggregator.

**Theorem 5.6.** $L_{HOI}(f_{uni}) = 3 - 2\sqrt{2}$.

Thus, surprisingly, the second-order information offers no help under the robust paradigm even if we assume experts are homogeneous. This motivates an additional natural assumption that we will introduce in the following section.

## 6 HOMOGENEOUS EXPERTS WITH NON-DEGENERATE SIGNALS

According to the proof of Theorem 5.5, when experts' recommended actions do not vary with their observed signals, the predictions contain no useful information and thus cannot aid the agent in achieving lower regret. In this section, alongside assuming expert homogeneity, we further assume the experts will recommend different actions when observing different signals, specifically that $b_L < 1/2 \leq b_H$.

**Assumption 6.1** (Homogeneous experts with non-degenerate signals). *Two experts are homogeneous. Also, they recommend different actions when observing different signals. In other words, $b_L < 1/2 \leq b_H$.*

This section studies the set of all possible conditionally independent information structures satisfying Assumption 6.1, denoted by NHI. Missing proofs of this section can be found in Appendix E.

## 6.1 Deterministic Aggregators

For deterministic aggregators, we notice that all results we establish in Section 5 still work for the information structure family NHI, with no changes in the proof. To summarize, we have the following corollaries.

**Corollary 6.2.** *For every deterministic aggregator* $f^d(a_1, a_2, p_1, p_2) \in F_{+2}$, $L_{\text{NHI}}(f^d) \geq 3 - 2\sqrt{2} \approx 0.1716$.

**Corollary 6.3.** $L_{\text{NHI}}(f_{thr}) = 1/3$.

**Corollary 6.4.** $L_{\text{NHI}}(f_{ftfe}) = 3 - 2\sqrt{2}$.

## 6.2 Random Aggregators

We first establish the lower bound for random aggregators in $\Delta(F_{+1})$. As with HOI, no random aggregator without second-order information can guarantee a regret less than $3 - 2\sqrt{2}$ for the worst case over NHI.

**Theorem 6.5.** *For every random aggregator* $f^r(a_1, a_2) \in \Delta(F_{+1})$, $L_{\text{NHI}}(f^r) \geq 3 - 2\sqrt{2}$.

Since the uniform aggregator guarantees a regret of $3 - 2\sqrt{2}$ for any information structure in HOI by Theorem 5.6, it guarantees at least this regret against all structures in NHI $\subset$ HOI. Thus, the aggregator is also optimal in this setting.

**Corollary 6.6.** $L_{\text{NHI}}(f_{uni}) = 3 - 2\sqrt{2}$.

We then come to consider aggregators utilizing second-order information, starting by establishing a lower bound, which is slightly smaller than that for random aggregators without second-order information.

**Theorem 6.7.** *For every random aggregator* $f^r(a_1, a_2, p_1, p_2) \in \Delta(F_{+2})$, $L_{\text{NHI}}(f^r) \geq 1/6 \approx 0.1667$.

To reduce the search space, we now study the characteristics of a robust random aggregator in $\Delta(F_{+2})$, aiming to achieve a low regret regarding information structures in NHI. First, we present a lemma showing that when two experts split in recommendation, the expert with recommendation 1 always has a no less prediction value than the other expert.

**Lemma 6.8.** *Suppose* $a_1 = 1$ *and* $a_2 = 0$, *then* $p_1 \geq p_2$.

Hence, it suffices for us to consider the scenario that $p_1 \geq p_2$ when $a_1 = 1$ and $a_2 = 0$. To add to this, we also have the following results:

**Proposition 6.9.** *There exists a random aggregator* $f^r$ *that achieves the lowest regret among all random aggregators in* $\Delta(F_{+2})$ *regarding* NHI *that satisfies the following for any* $a_1, a_2 \in \{0, 1\}$, $p_1, p_2, p \in [0, 1]$:

    *(a)* $f^r(1, 1, p_1, p_2) = 1$ *and* $f^r(0, 0, p_1, p_2) = 0$.
    *(b)* $f^r(a_1, a_2, p_1, p_2) = f^r(a_2, a_1, p_2, p_1)$.
    *(c)* $f^r(a_1, a_2, p_1, p_2) + f^r(1 - a_1, 1 - a_2, 1 - p_1, 1 - p_2) = 1$.
    *(d)* *when* $a_1 \neq a_2$, $f^r(a_1, a_2, p, 1 - p) = 0.5$.

The intuition behind (a) is that when the experts' recommendations agree, the agent straightforwardly takes that action. This follows directly from the information structure definition. (b) means

that the agent treats the two experts equally. (c) shows the equivalence of the two states. At last, (d) indicates that when two experts' predictions deviate from each other's true recommendations by the same amount, the aggregator shows no inclination toward either action. These three properties are proved by constructing another aggregator for any optimal one with the same regret guarantee, and then linearly combining them.

We now introduce a random aggregator, referred to as the "bipolar radial aggregator", that satisfies the criteria in Proposition 6.9. Moreover, $f_{bir}$ attains a lower regret over NHI compared to random aggregators without second-order information. This shows that predictive knowledge can help agents achieve better performance.

*The bipolar radial aggregator.* This aggregator follows the recommendation when the experts agree. When recommendations differ, it treats the experts equally and chooses based on how much their predictions deviate. Specifically, it fixes a center point $(0.6, 0.4)$ on the $p_1$-$p_2$ graph, outputs 0.5 around this center, and moves toward the extremes as the distance increases. This aggregator tends to trust the expert who predicts the other's action more accurately. Formally, when $a_1 = 1, a_2 = 0$, the aggregator is:

$$f_{bir}(1, 0, p_1, p_2) =$$
$$\begin{cases} \min\{1, (p_1 - 0.6)^2 + (p_2 - 0.4)^2 + 0.5\}, & p_1 + p_2 < 0.98 \\ \max\{0, 0.5 - (p_1 - 0.6)^2 - (p_2 - 0.4)^2\}, & p_1 + p_2 > 1.02 \\ 0.5, & \text{otherwise} \end{cases}$$

These parameters are set via experimentation. The case that $a_1 = 0, a_2 = 1$ is symmetric. We show the contour graph of the aggregator in the case of $a_1 = 1, a_2 = 0$ in Figure 1(a).

**Theorem 6.10.** $L_{\text{NHI}}(f_{bir}) \approx 0.1682$.

Theorem 6.10 leaves a gap between the upper and lower bound in the context of worst-case scenarios within NHI. Although closing this gap is challenging, we can enhance the upper bound by employing a more intricate aggregator derived from the algorithm introduced by Guo et al. [11], which views robust aggregation as a zero-sum game between nature and the aggregator and enables online learning techniques to solve the game effectively.

*The aggregator from the online learning algorithm.* As suggested by Proposition 6.9, this aggregator follows experts' recommendation when they agree. When they disagree, the aggregator treats the two experts equally and selects an action based on their predictions. Specifically, the algorithm learns an aggregator on discretized points $(p_1, p_2)$ where $p_1$ and $p_2$ are multiples of 0.1, and uses linear interpolation to give the output at other points. We present a contour graph of the algorithmic aggregator when $a_1 = 1$ and $a_2 = 0$ in Figure 1(b).

**Theorem 6.11.** $L_{\text{NHI}}(f_{alg}) \approx 0.1673$.

The bipolar radial aggregator in Figure 1(a) provides an intuitive and symmetrical way to weigh expert recommendations based on prediction accuracy and satisfies Proposition 6.9. The algorithmic aggregator in Figure 1(b) always favors action 0 in conflicts, which may seem unintuitive. However, its regret guarantee can be mirrored by an aggregator always favoring action 1 instead. Specifically, according to Proposition 6.9 and the linearity of the

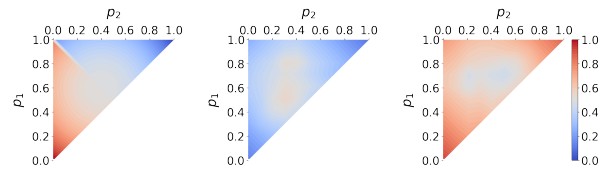

(a) The bipolar radial aggregator.  (b) The algorithmic aggregator.  (c) The mirror of the algorithmic aggregator.

**Figure 1: Contour graphs of different aggregators $f(1, 0, p_1, p_2)$ when the first expert recommends action $1$ and the second expert recommends action $0$. Colder to hotter shades represent the range of $f(1, 0, p_1, p_2)$ from $0$ to $1$. $f(0, 1, p_1, p_2) = f(1, 0, p_2, p_1)$. The region where $p_1 < p_2$ is not shown in the plot. This is because $p_1 < p_2$ is impossible under the assumption of homogeneous experts, as proved in Lemma 6.8. In practice, if we obtain reports $p_1 < p_2$, we simply pick an action uniformly at random.**

loss function, defining $f_{alg}^{\circ}$ as the mirror that flips predictions and outputs $f_{alg}^{\circ}(a_1, a_2, p_1, p_2) = 1 - f_{alg}(1 - a_2, 1 - a_1, 1 - p_2, 1 - p_1)$ guarantees the same regret $0.1673$. We present the contour graph of this aggregator when $a_1 = 1$ and $a_2 = 0$ in Figure 1(c). Therefore, to have a low regret, we can either favor action $0$ or favor action $1$ when the experts disagree, as long as the tendency varies with the predictions. Furthermore, we can observe a common pattern: medium values at the center and extreme values on both sides. These unexpected findings illustrate the intricacy of second-order information's role and complexity in small expert groups.

## 7 EXTENSION: GENERAL UTILITY FUNCTIONS WITH HOMOGENEOUS EXPERTS

This section extends the setting to general utility functions. We no longer assume that the agent's goal is to match the action with the correct state. Instead, we consider a more general scenario where the agent's utility is determined by both the state and the action taken, captured by a utility function $u : A \times \Omega \rightarrow R$.

Same as the original setting, the two experts will each recommend their preferred action $a_1, a_2$ according to the utility function and provide prediction $p_1, p_2$ about the probability of the other expert recommending action $1$. We further assume these two experts are homogeneous (Assumption 5.1). We still explore two layers of information: one where the agent can observe both recommended actions and predictions, and the other where the agent can only observe the recommended actions.

We also assume that $u(0, 0) > u(1, 0)$ and $u(1, 1) > u(0, 1)$, otherwise one action is dominated by another and the problem is trivial. Let the utility gap of two actions when the state is $0$ be $\Delta u_0 := u(0, 0) - u(1, 0)$ and the gap when the state is $1$ be $\Delta u_1 := u(1, 1) - u(0, 1)$. Also, we introduce their ratio as $t := \Delta u_1 / \Delta u_0$. Apparently, $t = 1$ in our original setting.

Note that the recommended action is still a threshold function of the posterior, parameterized by $t$, that is

$$a_i = \phi^t(b_i) = \mathbb{1}\left\{b_i \geq \frac{u(0, 0) - u(1, 0)}{u(0, 0) - u(1, 0) + u(1, 1) - u(0, 1)}\right\}$$

$$= \mathbb{1}\left\{b_i \geq \frac{1}{t + 1}\right\}.$$

The action of the benchmark when observing signals $(s_1, s_2)$ regarding information structure $\pi$ should be $a^* := \phi^t(\pi(\omega = 1 \mid S_1 = s_1, S_2 = s_2))$.

We focus on random aggregators. The regret of any random aggregator $f^r$ with second-order information regarding information structure $\pi$ can be defined as

$$L(f^r, \pi, u) := \mathbb{E}_{\pi, f^r(\cdot)}[u(a^*, \omega) - u(f^r(a_1, a_2, p_1, p_2), \omega)].$$

The regret of any random aggregator without second-order information is similar:

$$L(f^r, \pi, u) := \mathbb{E}_{\pi, f^r(\cdot)}[u(a^*, \omega) - u(f^r(a_1, a_2), \omega)].$$

Here the randomness comes from the information structure $\pi$ and the output of random aggregator $f^r$.

Building upon the preceding results, we start by establishing negative outcomes for possibly degenerate signals. We then focus on non-degenerate signals as Section 6 and introduce robust aggregators with second-order information, tailored to different utility ratios, demonstrating that second-order information empowers the agent to achieve lower regret across many utility functions. Due to the space limit, we defer the details to Appendix A.

## 8 CONCLUSION AND DISCUSSION

In this work, we study the benefit of second-order information in decision aggregation with two experts. Specifically, we investigate binary actions, binary states, and binary signals, examining the optimal deterministic and random aggregators, both with and without second-order information. Our research provides insight into the crucial question about the effectiveness of second-order information in mitigating regret, with various underlying assumptions. Furthermore, we extend our findings to encompass more general utility functions, thereby broadening our results' applicability scope. Future research directions of this work include conducting real-world experiments, exploring the scenario with more experts, and considering more complex signal settings.

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

# A DETAILS IN SECTION 7 – GENERAL UTILITY FUNCTIONS WITH HOMOGENEOUS EXPERTS

We follow the framework in Section 7. Now, we consider a special utility function that satisfies $u^t(0,1) = u^t(1,0) = 0$, $u^t(0,0) = 1$, and $u^t(1,1) = t$, and define the regret function facing this utility function:

$$L^t(f^r, \pi) := \mathbb{E}_{\pi, f^r(\cdot)}[u^t(a^*, \omega) - u^t(f^r(a_1, a_2, p_1, p_2), \omega)].$$

$$L^t(f^r, \pi) := \mathbb{E}_{\pi, f^r(\cdot)}[u^t(a^*, \omega) - u^t(f^r(a_1, a_2), \omega)].$$

We then show that for all general utility functions with the utility gap ratio $t$, the problem is identical to the one with utility function $u^t$.

**Proposition A.1.** *For any utility function $u$ with the utility gap $\Delta u_1, \Delta u_0$ and ratio $t$, $L(f^r, \pi, u) = \Delta u_0 \cdot L^t(f^r, \pi)$ holds for any random aggregator $f^r$ and information structure $\pi$.*

Therefore, we only consider the utility functions of this special utility family in the rest of this section. Under the utility function $u^t$, the regret of random aggregator $f^r$ regarding information structure family $P$ is defined as:

$$L_P^t(f^r) := \max_{\pi \in P} L^t(f^r, \pi).$$

As suggested, we focus on random aggregators in this part. Missing proofs of this section can be found in Appendix F.

## A.1 A Negative Result: Predictions are Useless in General

We begin by presenting the lower bound for the regret of any random aggregator equipped with second-order information. Notably, the expressions differ slightly depending on whether $t$ is greater than 1 or less than 1.

**Theorem A.2.** *For any $t \geq 1$ and random aggregator $f^r(a_1, a_2, p_1, p_2) \in \Delta(F_{+2})$, $L_{\text{HOI}}^t(f^r) \geq (\sqrt{1 + 1/t} - \sqrt{1/t})^2$.*

**Theorem A.3.** *For any $t \leq 1$ and random aggregator $f^r(a_1, a_2, p_1, p_2) \in \Delta(F_{+2})$, $L_{\text{HOI}}^t(f^r) \geq (\sqrt{t + t^2} - t)^2$.*

The proofs of Theorems A.2 and A.3 can be easily extended from the proof of Theorem 5.5, so we omit them here and defer them to Appendices F.2 and F.3.

*The prob-$p$ aggregator.* The prob-$p$ aggregator follows the recommendations of experts when they agree. In cases where they disagree, the aggregator selects action 1 with probability $p$ and action 0 with probability $1 - p$. In mathematical terms, the prob-$p$ aggregator can be expressed as

$$f_p(a_1, a_2) = \begin{cases} a_1 & a_1 = a_2 \\ p & a_1 \neq a_2 \end{cases}.$$

Also, this aggregator can be seen as a generalization of the uniform aggregator ($p = 0.5$) we introduced earlier. We have the following two positive results.

**Theorem A.4.** *For any $t \geq 1$, and $p \in [0.5, (\sqrt{1 + 1/t} - \sqrt{1/t})^2/(2(\sqrt{t + t^2} - t)^2)]$, $L_{\text{HOI}}^t(f_p) = (\sqrt{1 + 1/t} - \sqrt{1/t})^2$.*

**Theorem A.5.** *For any $t \leq 1$, and $p \in [1 - (\sqrt{t + t^2} - t)^2/(2(\sqrt{1 + 1/t} - \sqrt{1/t})^2), 0.5]$, $L_{\text{HOI}}^t(f_p) = (\sqrt{t + t^2} - t)^2$.*

The techniques used to analyze the aggregator's regret in Theorems A.4 and A.5 parallel those employed in the proof of Theorem 5.6. These are deferred to Appendices F.4 and F.5. We also notice from the above two theorems that the uniform aggregator ($p = 0.5$) guarantees the lowest regret among aggregators without second-order information for any utility function.

## A.2 Non-Degenerate Signals: Predictions are Useful

Similarly, following the exploration in Section 6, we now extend our analysis to encompass non-degenerate signals. We begin by establishing a lower bound for random aggregators lacking second-order information.

**Theorem A.6.** *For every random aggregator $f^r(a_1, a_2) \in \Delta(F_{+1})$,*

$$L_{\text{NHI}}(f^r) \geq \frac{2\left(\sqrt{1 + 1/t} - \sqrt{1/t}\right)^2 (\sqrt{t + t^2} - t)^2}{(\sqrt{t + t^2} - t)^2 + \left(\sqrt{1 + 1/t} - \sqrt{1/t}\right)^2}.$$

We proceed to demonstrate that for any utility function, there exists a specific value of $p$ such that the prob-$p$ aggregator attains the lowest regret among all aggregators without second-order information in NHI.

**Theorem A.7.** *For any $t$,*

$$L_{\text{NHI}}^t(f_p) = \frac{2\left(\sqrt{1 + 1/t} - \sqrt{1/t}\right)^2 (\sqrt{t + t^2} - t)^2}{(\sqrt{t + t^2} - t)^2 + \left(\sqrt{1 + 1/t} - \sqrt{1/t}\right)^2}$$

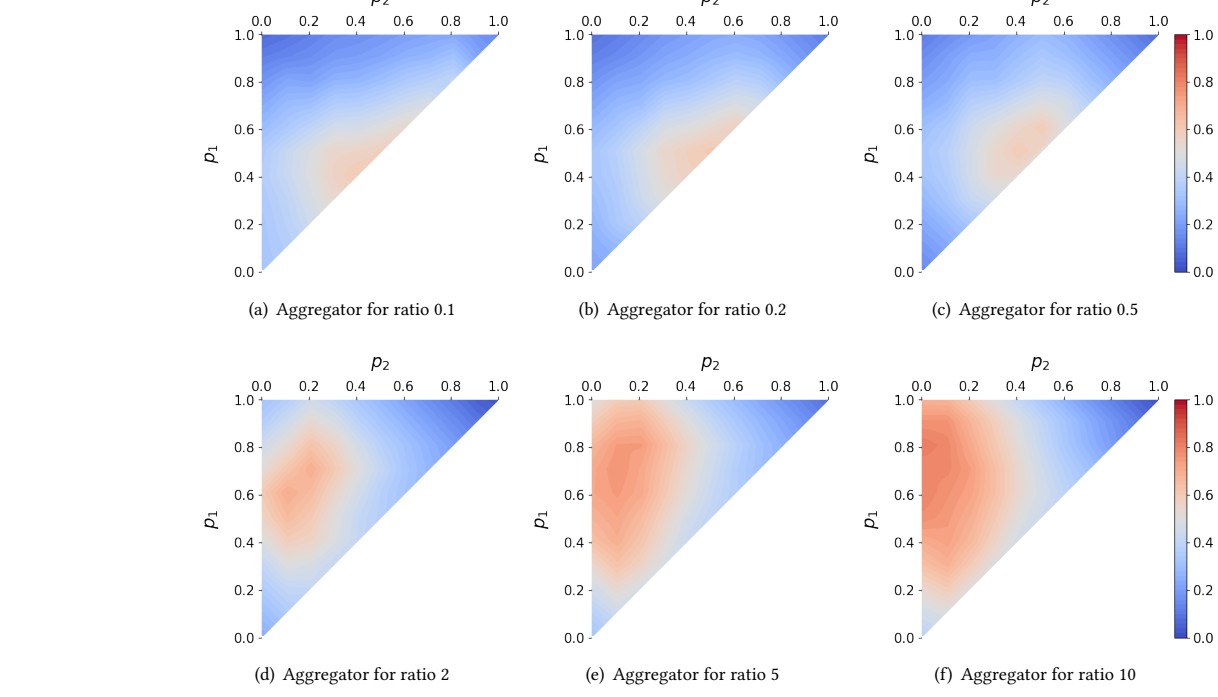

(a) Aggregator for ratio 0.1          (b) Aggregator for ratio 0.2          (c) Aggregator for ratio 0.5

(d) Aggregator for ratio 2          (e) Aggregator for ratio 5          (f) Aggregator for ratio 10

**Figure 2: Contour graphs of different aggregators for general utility functions when the first expert recommends $1$ and the second expert recommends $0$. Colder to hotter shades represent the range of $f(1, 0, p_1, p_2)$ from $0$ to $1$. $f(0, 1, p_1, p_2) = f(1, 0, p_2, p_1)$.**

**Table 2: Comparison between the regrets of the best aggregators without second-order information and our aggregators with second-order information.**

| Ratio | Regret of aggregators | |
|---|---|---|
| | In Theorem A.7 | In Figure 2 |
| 0.1 | 0.0330 | 0.0233 |
| 0.2 | 0.0591 | 0.0432 |
| 0.5 | 0.1152 | 0.0956 |
| 1 | 0.1716 | 0.1673 |
| 2 | 0.2304 | 0.1927 |
| 5 | 0.2954 | 0.2157 |
| 10 | 0.3300 | 0.2293 |

*for*

$$p = \frac{\left(\sqrt{1 + 1/t} - \sqrt{1/t}\right)^2}{(\sqrt{t + t^2} - t)^2 + \left(\sqrt{1 + 1/t} - \sqrt{1/t}\right)^2}.$$

Regarding aggregators with second-order information, we demonstrate that they are capable of achieving a lower regret, akin to the scenario of $t = 1$. On this side, we present a series of aggregators derived from the online learning algorithm discussed in Guo et al. [11] for varying $t$ in Figure 2. To analyze their regrets, we use the same method as we prove Theorem 6.11 and present the results in Table 2. Here, note that Lemma 6.8 and Proposition 6.9(a)(b) still work for general functions; thus we only draw the upper-left triangle cases with $a_1 = 1$, $a_2 = 0$, and $p_1 \geq p_2$. The case of $f(0, 1, p_1, p_2)$ with $p_2 \geq p_1$ is symmetric.

Our findings reveal that, across all six ratios, aggregators utilizing second-order information exhibit significant improvements over the best aggregators without this additional information. Furthermore, similar to the aggregator for ratio 1 in Figure 1(b), their output distribution still spans the range between 0 and a value below 1, and this upper bound increases as the ratio grows. Additionally, the point with the

highest output value gradually moves from the center toward the edge that corresponds to the expert with less prediction accuracy when recommending 0.

We should also notice that when $t \geq 1$, the regret of any aggregator facing the ratio $t$ instance is $t$ times the regret of its mirror (defined in Section 6) facing the ratio $1/t$ instance. Therefore, by Proposition A.1 given any robust aggregator against ratio $t$, we can naturally construct a robust aggregator against the ratio $1/t$. These aggregators collectively highlight the potential of second-order information in reducing regret in worst-case scenarios within NHI.

## B MISSING PROOFS IN SECTION 3

### B.1 Proof of Theorem 3.1

By Lemma 3.2, to bound the expected regret of any random aggregator in $\Delta(F_{+2})$, it suffices to analyze the expected regret of the deterministic aggregator in $F_{+2}$ relative to the same benchmark, where information structures are drawn from a carefully selected distribution. We now give a distribution $D \in \Delta(\text{ALL})$ over two information structures, regarding which any aggregator in $F_{+2}$ cannot achieve a regret below $1/2$. These two information structures are as follows:

| $\mu_1 = 1/2$ | $\pi_1(s_1, s_2 \mid \omega = 1)$ | $\pi_1(s_1, s_2 \mid \omega = 0)$ | $\mu_2 = 1/2$ | $\pi_2(s_1, s_2 \mid \omega = 1)$ | $\pi_2(s_1, s_2 \mid \omega = 0)$ |
|---|---|---|---|---|---|
| $(L, L)$ | $0$ | $1/2 + \epsilon$ | $(L, L)$ | $1/2 - \epsilon$ | $2\epsilon$ |
| $(L, H)$ | $1/2 - \epsilon$ | $0$ | $(L, H)$ | $0$ | $1/2 - \epsilon$ |
| $(H, L)$ | $1/2 - \epsilon$ | $0$ | $(H, L)$ | $0$ | $1/2 - \epsilon$ |
| $(H, H)$ | $2\epsilon$ | $1/2 - \epsilon$ | $(H, H)$ | $1/2 + \epsilon$ | $0$ |

Note that each of them has a marginal distribution

$$\mu = \frac{1}{2}, \pi(s = L \mid \omega = 1) = \frac{1}{2} - \epsilon, \pi(s = H \mid \omega = 0) = \frac{1}{2} + \epsilon.$$

Therefore, when observing signal $L$, the expert will recommend action 0; when observing signal $H$, the expert will recommend action 1. In the following discussion, we consider $\epsilon$ as a small positive number less than $1/2$.

In the first information structure, two experts always see the same signal when the real state is 0, which means $(H, H)$ happens with probability $1/2 - \epsilon$ and $(L, L)$ happens with probability $1/2 + \epsilon$. When the real state is 1, two experts see different signals most of the time. In detail, $(H, L), (L, H)$ each happens with probability $1/2 - \epsilon$ and $(H, H)$ happens with probability $2\epsilon$.

In the second information structure, $(H, H)$ happens with probability $1/2 + \epsilon$, and $(L, L)$ happens with probability $1/2 - \epsilon$ when the real state is 1. When the real state is 0, $(H, L), (L, H)$ each happens with probability $1/2 - \epsilon$, and $(L, L)$ happens with probability $2\epsilon$.

Notice that inputs $(1, 0, 1/2 + \epsilon, 1/2 - \epsilon)$ and $(0, 1, 1/2 - \epsilon, 1/2 + \epsilon)$ each happens with probability $1/4 - \epsilon/2$ in both information structures. However, the real state behind these inputs is 1 in the first information structure but 0 in the second. Also, inputs $(1, 1, 1/2 + \epsilon, 1/2 + \epsilon)$ and $(0, 0, 1/2 - \epsilon, 1/2 - \epsilon)$ are with real state 0 most of the time in the first information structure and with real state 1 most of the time in the second one.

Now consider the distribution that the real information structure can be the first or the second one with equal probability. The optimal aggregator in $F_{+2}$ is independent of the specific outputs generated by $(1, 0, 1/2 + \epsilon, 1/2 - \epsilon)$ and $(0, 1, 1/2 - \epsilon, 1/2 + \epsilon)$. Also, the optimal aggregator outputs 1 for $(1, 1, 1/2 + \epsilon, 1/2 + \epsilon)$ and outputs 0 for $(1, 1, 1/2 + \epsilon, 1/2 + \epsilon)$. However, the benchmark can always identify the more possible state according to the knowledge of the real information structure. Thus, the relative regret of any aggregator in $F_{+2}$ regarding this distribution of information structures is at least

$$\frac{1}{2} \times 2 \times (\frac{1}{4} - \frac{\epsilon}{2}) \times (1 - 0) + 2 \times \frac{1}{2} \times (\frac{1}{4} - \frac{\epsilon}{2} - \frac{\epsilon}{2}) \times (1 - 0) = \frac{1}{2} - \frac{3\epsilon}{2}.$$

Since $\epsilon$ can be arbitrarily small, the theorem holds.

### B.2 Proof of Theorem 3.3

Since the first expert's probability of recommending a bad action is $\min\{\mu, 1 - \mu\}$ without additional information, and this probability decreases when the signal provides useful state information, the aggregator's regret from always following the first expert is bounded above by $\min\{\mu, 1 - \mu\} \leq 1/2$. Combining this with the lower bound from Theorem 3.1 establishes that $L_{\text{ALL}}(f_{ftfe}) = 1/2$.

## C MISSING PROOFS IN SECTION 4

### C.1 Proof of Theorem 4.1

We now present two information structures in CI and demonstrate that for any deterministic aggregator in $F_{+1}$, there exists an input that will inevitably result in a regret of 0.5 for one of these information structures, regardless of the output of the deterministic aggregator for that input. These two information structures are as follows:

| $\mu_1 = 0.5 + \epsilon$ | $\pi_1(s = L \mid \omega = 1)$ | $\pi_1(s = L \mid \omega = 0)$ |
| --- | --- | --- |
| Expert 1 | 0.5 | 0.5 |
| Expert 2 | 0 | 1 |

| $\mu_2 = 0.5 - \epsilon$ | $\pi_2(s = L \mid \omega = 1)$ | $\pi_2(s = L \mid \omega = 0)$ |
| --- | --- | --- |
| Expert 1 | 0 | 1 |
| Expert 2 | 0.5 | 0.5 |

For the first information structure, we set $(\mu, k_1, k_2, l_1, l_2) = (0.5 + \epsilon, 0.5, 0, 0.5, 1)$, and the posterior is $(b_{1L}, b_{2L}, b_{1H}, b_{2H}) = (0.5 + \epsilon, 0, 0.5 + \epsilon, 1)$. Here, we suppose $\epsilon$ is a small positive number less than 0.5. In this information structure, the first expert has no information beyond the prior, and thus will always recommend action 1 regardless of the observed signal. On the other hand, the second expert possesses complete knowledge of the true state.

The second information structure is symmetric to the first one, with $(\mu, k_1, k_2, l_1, l_2) = (0.5 - \epsilon, 0, 0.5, 1, 0.5)$ and posterior $(b_{1L}, b_{2L}, b_{1H}, b_{2H}) = (0, 0.5 - \epsilon, 1, 0.5 - \epsilon)$. In this scenario, the first expert is omniscient, while the second expert has no information beyond the prior and always recommends action 0.

Notice that the input $(1, 0)$ happens with probability $0.5 - \epsilon$ in both information structures. However, the real state is 1 in the first information structure and 0 in the second one with input $(1, 0)$. Also, the benchmark always knows the real state from the knowledge of information structure. Therefore, whatever the deterministic aggregator outputs for this input, it always causes a regret of $0.5 - \epsilon$ regarding one of the information structures. Since $\epsilon$ can be arbitrarily small, we derive the theorem.

## C.2 Proof of Theorem 4.3

We, again, present two information structures in CI and demonstrate that for any deterministic aggregator in $F_{+2}$, there exists an input that will inevitably result in a regret of 1/3 for one of these information structures.

| $\mu_1 = (1 + 2\epsilon)/(3 + 2\epsilon)$ | $\pi_1(s = L \mid \omega = 1)$ | $\pi_1(s = L \mid \omega = 0)$ |
| --- | --- | --- |
| Expert 1 | 0 | $0.5 + \epsilon$ |
| Expert 2 | 0 | 1 |

| $\mu_2 = 2/(3 + 2\epsilon)$ | $\pi_2(s = L \mid \omega = 1)$ | $\pi_2(s = L \mid \omega = 0)$ |
| --- | --- | --- |
| Expert 1 | 0 | 1 |
| Expert 2 | $0.5 - \epsilon$ | 1 |

For the first information structure, we set $(\mu, k_1, k_2, l_1, l_2) = ((1 + 2\epsilon)/(3 + 2\epsilon), 0, 0, 0.5 + \epsilon, 1)$, and the posterior is $(b_{1L}, b_{2L}, b_{1H}, b_{2H}) = (0, 0, 0.5 + \epsilon, 1)$. We suppose $\epsilon \to 0^+$. In this information structure, when observing signal $L$, the first expert possesses complete certainty regarding the state. However, for small $\epsilon$, its level of certainty diminishes significantly when observing signal $H$. On the other hand, the second expert is omniscient.

We construct a symmetric one for the second information structure to the first. Now $(\mu, k_1, k_2, l_1, l_2) = (2/(3 + 2\epsilon), 0, 0.5 - \epsilon, 1, 1)$, which leads to the posterior $(b_{1L}, b_{2L}, b_{1H}, b_{2H}) = (0.5 - \epsilon, 0, 1, 1)$. In this information structure, however, the first expert is always sure about the state. On the other hand, the second expert possesses complete certainty regarding the state when the realized signal is $H$. However, for small $\epsilon$, its level of certainty diminishes significantly when observing signal $L$.

Notice that the input $(1, 0, 0.5 + \epsilon, 0.5 - \epsilon)$ happens with $(1 - 2\epsilon)/(3 + 2\epsilon)$ in both information structures. However, the real state is 0 in the first information structure and 1 in the second one. Also, the benchmark always knows the real state by identifying the omniscient expert. Therefore, whatever the deterministic aggregator outputs for this input, there should be a regret of $(1 - 2\epsilon)/(3 + 2\epsilon)$ regarding one of the information structures. We finish the proof of the theorem when $\epsilon \to 0^+$.

## C.3 Proof of Theorem 4.4

Before proving the theorem, we give a lemma to demonstrate specific numeric relationships for information structure parameters, contributing to the proof's brevity.

**Lemma C.1.** *For any information structure in CI, $k_i \leq l_i$ for $i = 1, 2$.*

PROOF OF LEMMA C.1. To prove the lemma, we first recall the key parameters of an information structure as follows:

$$\mu = \pi(\omega = 1);$$
$$k_1 = \pi(S_1 = L \mid \omega = 1), \quad l_1 = \pi(S_1 = L \mid \omega = 0);$$
$$k_2 = \pi(S_2 = L \mid \omega = 1), \quad l_2 = \pi(S_2 = L \mid \omega = 0).$$

Now, by our notations, $\pi(\omega = 1 \mid S_i = L) \leq \mu \leq \pi(\omega = 1 \mid S_i = H)$ for both $i = 1, 2$. Under the above notations, that is to say

$$\frac{\mu k_i}{\mu k_i + (1 - \mu)l_i} \leq \mu \leq \frac{\mu(1 - k_i)}{\mu(1 - k_i) + (1 - \mu)(1 - l_i)} \iff k_i \leq l_i, \quad \forall i = 1, 2.$$

$\square$

We come back to our proof. Recall the definition of regret for any information structure $\pi$:

$$L(f, \pi) = \sum_{\omega \in \Omega, s_1, s_2 \in S} \pi(\omega, s_1, s_2)(u(\phi(\pi(\omega = 1 \mid s_1, s_2)), \omega) - f(a_1(s_1), a_2(s_2), p_1(s_1), p_2(s_2))u(1, \omega)$$
$$- (1 - f(a_1(s_1), a_2(s_2), p_1(s_1), p_2(s_2)))u(0, \omega)).$$

To compute the maximum regret of $f_{thr}$, we now consider all possible seven different cases of $\pi$. Here, notice that $k_1 \leq l_1$ and $k_2 \leq l_2$ hold due to Lemma C.1. In each of the cases, the corresponding program with strictly bounded Lipschitz continuity is solved by Wolfram Mathematica. This is the same for other positive results with similar proof.

*Case 1: $\mu k_1 > (1-\mu)l_1, \mu k_2 > (1-\mu)l_2$.* In this case, we have $a_i(s) = 1$ for all $i = 1, 2$ and $s = L, H$, and the aggregator always chooses action 1. Meanwhile, we notice in this case that

$$\pi(\omega = 1 \mid S_1 = L, S_2 = H) = \frac{\mu k_1(1-k_2)}{\mu k_1(1-k_2) + (1-\mu)l_1(1-l_2)} \geq \frac{1}{2},$$

$$\pi(\omega = 1 \mid S_1 = H, S_2 = L) = \frac{\mu(1-k_1)k_2}{\mu(1-k_1)k_2 + (1-\mu)(1-l_1)l_2} \geq \frac{1}{2},$$

$$\pi(\omega = 1 \mid S_1 = H, S_2 = H) = \frac{\mu(1-k_1)(1-k_2)}{\mu(1-k_1)(1-k_2) + (1-\mu)(1-l_1)(1-l_2)} \geq \frac{1}{2}.$$

Therefore, the only possible difference between the threshold aggregator and the benchmark is under the condition that $S_1 = S_2 = L$, and the regret is bounded by the following program:

$$\begin{aligned}
\max \quad & -\mu k_1 k_2 + (1-\mu)l_1 l_2, \\
\text{s.t.} \quad & \mu k_1 k_2 \leq (1-\mu)l_1 l_2, \\
& \mu k_1 \geq (1-\mu)l_1, \mu k_2 \geq (1-\mu)l_2, \\
& 0 \leq k_1 \leq l_1 \leq 1, 0 \leq k_2 \leq l_2 \leq 1, 0 \leq \mu \leq 1.
\end{aligned}$$

And the optimum of the above is $3 - 2\sqrt{2} \approx 0.1716$, which is reached when $\mu = \sqrt{2}/2, k_1 = k_2 = \sqrt{2} - 1, l_1 = l_2 = 1$.

*Case 2: $\mu(1-k_1) < (1-\mu)(1-l_1), \mu(1-k_2) < (1-\mu)(1-l_2)$.* This case is equivalent to Case 1 by substituting $\mu$ with $1 - \mu$ and $k_i$ with $1 - l_i$ for $i = 1, 2$. Thus, they have the same optimum $3 - 2\sqrt{2}$.

*Case 3: $\mu k_1 \leq (1-\mu)l_1, \mu(1-k_1) \geq (1-\mu)(1-l_1), \mu(1-k_2) < (1-\mu)(1-l_2)$.* In this case, we have $a_1(L) = 0, a_1(H) = 1$, and $a_2(L) = a_2(H) = 0$. We also observe that

$$\pi(\omega = 1 \mid S_1 = L, S_2 = L) = \frac{\mu k_1 k_2}{\mu k_1 k_2 + (1-\mu)l_1 l_2} \leq \frac{1}{2}.$$

$$\pi(\omega = 1 \mid S_1 = L, S_2 = H) = \frac{\mu k_1(1-k_2)}{\mu k_1(1-k_2) + (1-\mu)l_1(1-l_2)} \leq \frac{1}{2}.$$

Thus, the threshold aggregator matches the benchmark when $S_1 = L$. On the other hand, when $S_1 = H$, there is a split between two experts, with peer predictions $p_1 = 0, p_2 < 1$. Hence, our threshold aggregator would choose action 1. We further have

$$\pi(\omega = 1 \mid S_1 = H, S_2 = H) = \frac{\mu(1-k_1)(1-k_2)}{\mu(1-k_1)(1-k_2) + (1-\mu)(1-l_1)(1-l_2)} \geq \frac{1}{2},$$

As a result, the regret only comes from the circumstance that $S_1 = H, S_2 = L$, and the regret is bounded by the following program:

$$\begin{aligned}
\max \quad & -\mu(1-k_1)k_2 + (1-\mu)(1-l_1)l_2, \\
\text{s.t.} \quad & \mu(1-k_1)k_2 \leq (1-\mu)(1-l_1)l_2, \\
& \mu k_1 \leq (1-\mu)l_1, \mu(1-k_1) \geq (1-\mu)(1-l_1), \mu(1-k_2) \leq (1-\mu)(1-l_2), \\
& 0 \leq k_1 \leq l_1 \leq 1, 0 \leq k_2 \leq l_2 \leq 1, 0 \leq \mu \leq 1.
\end{aligned}$$

The above program has a maximum value of $3 - 2\sqrt{2}$ when $\mu = 1 - \sqrt{2}/2, k_1 = k_2 = 0, l_1 = l_2 = 2 - \sqrt{2}$.

*Case 4: $\mu k_1 \leq (1-\mu)l_1, \mu(1-k_1) \geq (1-\mu)(1-l_1), \mu k_2 > (1-\mu)l_2$.* This case is equivalent to Case 3 only by substituting $\mu$ with $1 - \mu$, $k_i$ with $1 - l_i$ for $i = 1, 2$, and they have the same optimum $3 - 2\sqrt{2}$.

*Case 5: $\mu(1-k_1) < (1-\mu)(1-l_1), \mu k_2 \leq (1-\mu)l_2, \mu(1-k_2) \geq (1-\mu)(1-l_2)$.* This case is equivalent to Case 3 by swapping the order of two experts, and we omit it.

*Case 6: $\mu k_1 > (1-\mu)l_1, \mu k_2 \leq (1-\mu)l_2, \mu(1-k_2) \geq (1-\mu)(1-l_2)$.* This case is equivalent to Case 4 by swapping the order of two experts, and we omit it.

*Case 7:* $\mu k_1 \leq (1-\mu)l_1, \mu(1-k_1) \geq (1-\mu)(1-l_1), \mu k_2 \leq (1-\mu)l_2, \mu(1-k_2) \geq (1-\mu)(1-l_2)$. This case is the most complex one. We have $a_1(L) = a_2(L) = 0$ and $a_1(H) = a_2(H) = 1$. Meanwhile,

$$\pi(\omega = 1 \mid S_1 = L, S_2 = L) = \frac{\mu k_1 k_2}{\mu k_1 k_2 + (1-\mu)l_1 l_2} \leq \frac{1}{2},$$

$$\pi(\omega = 1 \mid S_1 = H, S_2 = H) = \frac{\mu(1-k_1)(1-k_2)}{\mu(1-k_1)(1-k_2) + (1-\mu)(1-l_1)(1-l_2)} \geq \frac{1}{2}.$$

Therefore, it is guaranteed that the threshold aggregator is identical to the benchmark when $S_1 = S_2$. We now consider the case when $S_1 \neq S_2$, i.e., there is a split between the two experts. We now formalize their predictions:

$$p_1^+ := \pi(a_2 = 1 \mid S_1 = L) = \pi(S_2 = H \mid S_1 = L) = \frac{\mu k_1(1-k_2) + (1-\mu)l_1(1-l_2)}{\mu k_1 + (1-\mu)l_1},$$

$$p_1^- := \pi(a_2 = 1 \mid S_1 = H) = \frac{\mu(1-k_1)(1-k_2) + (1-\mu)(1-l_1)(1-l_2)}{\mu(1-k_1) + (1-\mu)(1-l_1)},$$

$$p_2^+ := \pi(a_1 = 1 \mid S_2 = H) = \frac{\mu(1-k_1)(1-k_2) + (1-\mu)(1-l_1)(1-l_2)}{\mu(1-k_2) + (1-\mu)(1-l_2)},$$

$$p_2^- := \pi(a_1 = 1 \mid S_2 = L) = \frac{\mu(1-k_1)k_2 + (1-\mu)(1-l_1)l_2}{\mu k_2 + (1-\mu)l_2}.$$

Now, when $p_1^+ + p_2^+ > 1$ and $p_1^- + p_2^- > 1$ both hold, the threshold aggregator outputs action 0 when two experts split, and the regret is bounded in this sub-case by

$$\max \quad (\mu k_1(1-k_2) - (1-\mu)l_1(1-l_2))^+ + (\mu(1-k_1)k_2 - (1-\mu)(1-l_1)l_2)^+,$$

$$\text{s.t.} \quad p_1^+ + p_2^+ \geq 1, p_1^- + p_2^- \geq 1,$$

$$\mu k_1 \leq (1-\mu)l_1, \mu(1-k_1) \geq (1-\mu)(1-l_1),$$

$$\mu k_2 \leq (1-\mu)l_2, \mu(1-k_2) \geq (1-\mu)(1-l_2),$$

$$0 \leq k_1 \leq l_1 \leq 1, 0 \leq k_2 \leq l_2 \leq 1, 0 \leq \mu \leq 1.$$

The above reaches the optimum of $1/3$ when $\mu = 2/3, k_1 = 0, k_2 = 0.5, l_1 = l_2 = 1$ or $\mu = 3/4, k_1 = k_2 = 1/3, l_1 = l_2 = 1$.

Conversely, when $p_1^+ + p_2^+ \leq 1$ and $p_1^- + p_2^- \leq 1$ both hold, the threshold aggregator outputs action 1 with split decisions, and the regret is bounded by

$$\max \quad (-\mu k_1(1-k_2) + (1-\mu)l_1(1-l_2))^+ + (-\mu(1-k_1)k_2 + (1-\mu)(1-l_1)l_2)^+,$$

$$\text{s.t.} \quad p_1^+ + p_2^+ \leq 1, p_1^- + p_2^- \leq 1,$$

$$\mu k_1 \leq (1-\mu)l_1, \mu(1-k_1) \geq (1-\mu)(1-l_1),$$

$$\mu k_2 \leq (1-\mu)l_2, \mu(1-k_2) \geq (1-\mu)(1-l_2),$$

$$0 \leq k_1 \leq l_1 \leq 1, 0 \leq k_2 \leq l_2 \leq 1, 0 \leq \mu \leq 1.$$

This program has a maximum of $1/3$ when $\mu = 1/4, k_1 = k_2 = 0, l_1 = l_2 = 2/3$ or $\mu = 1/3, k_1 = k_2 = 0, l_1 = 1, l_2 = 0.5$.

For the rest cases, we derive by symbolic computation that:

$$p_1^+ + p_2^+ - 1 = \frac{(\mu k_1(1-k_2) + (1-\mu)l_1(1-l_2))((1-l_1-l_2)(1-\mu) + (1-k_1-k_2)\mu)}{(\mu k_1 + (1-\mu)l_1)(\mu(1-k_2) + (1-\mu)(1-l_2))},$$

$$p_1^- + p_2^- - 1 = \frac{(\mu(1-k_1)k_2 + (1-\mu)(1-l_1)l_2)((1-l_1-l_2)(1-\mu) + (1-k_1-k_2)\mu)}{(\mu(1-k_1) + (1-\mu)(1-l_1))(\mu k_2 + (1-\mu)l_2)}.$$

Thus, we know that $(p_1^+ + p_2^+ - 1)(p_1^- + p_2^- - 1) \geq 0$, and we are only left with the cases that $p_1^+ + p_2^+ = 1, p_1^- + p_2^- > 1$ and $p_1^+ + p_2^+ > 1, p_1^- + p_2^- = 1$, which are symmetric. We consider the first case, under which our algorithm outputs action 1 when $S_1 = L$ and $S_2 = H$, and outputs action 0 when $S_1 = H$ and $S_2 = L$. We have

$$p_1^+ + p_2^+ = 1 \iff (l_1(1-l_2)(1-\mu) + k_1(1-k_2)\mu)((1-l_1-l_2)(1-\mu) + (1-k_1-k_2)\mu) = 0.$$

Now that $0 < \mu < 1$, we have $l_1 > 0$ and $k_2 < 1$, and there are two remaining possibilities:

1. $k_1 = 0, l_2 = 1$. In this case, the program's upper bound becomes:

$$\max \quad (\mu k_2 - (1 - \mu)(1 - l_1))^+,$$
$$\text{s.t.} \quad p_1^- + p_2^- \geq 1,$$
$$\mu \geq (1 - \mu)(1 - l_1), \mu k_2 \leq 1 - \mu,$$
$$0 \leq l_1 \leq 1, 0 \leq k_2 \leq 1, 0 \leq \mu \leq 1.$$

And the optimum value is $1/3$, reached at $\mu = 2/3, k_2 = 0.5, l_1 = 1$.

2. $(1 - l_1 - l_2)(1 - \mu) + (1 - k_1 - k_2)\mu = 0$. Under this condition, we achieve that $p_1^- + p_2^- = 1$ holds as well, which is a contradiction.

Thus, we conclude for Case 7 that the maximum regret is $1/3$.

Synthesizing all 7 cases, we achieve that $L(f_{thr}) \leq 1/3$. Combining with the lower bound in Theorem 4.3, we finish the proof of $L(f_{thr}) = 1/3$.

## C.4 Proof of Theorem 4.5

We also prove by Lemma 3.2. To bound the expected regret of any random aggregator in $\Delta(F_{+2})$, it suffices to analyze the expected regret of the deterministic aggregator in $F_{+2}$ relative to the same benchmark, where information structures are drawn from a carefully selected distribution.

We then give a distribution $D \in \Delta(CI)$ over two conditionally independent information structures, regarding which any aggregator in $F_{+2}$ cannot achieve a regret below $1/4$.

| $\mu_1 = 0.5$ | $\pi_1(s = L \mid \omega = 1)$ | $\pi_1(s = L \mid \omega = 0)$ | $\mu_2 = 0.5$ | $\pi_2(s = L \mid \omega = 1)$ | $\pi_2(s = L \mid \omega = 0)$ |
|---|---|---|---|---|---|
| Expert 1 | 0 | 1 | Expert 1 | $0.5 - \epsilon$ | $0.5 + \epsilon$ |
| Expert 2 | $0.5 - \epsilon$ | $0.5 + \epsilon$ | Expert 2 | 0 | 1 |

The first information structure has parameters $(\mu, k_1, k_2, l_1, l_2) = (0.5, 0, 0.5 - \epsilon, 1, 0.5 + \epsilon)$ and leads to the posterior $(b_{1L}, b_{2L}, b_{1H}, b_{2H}) = (0, 0.5 - \epsilon, 1, 0.5 + \epsilon)$. Again, we suppose $\epsilon$ is a small positive number that is less than 0.5. In this information structure, the first expert is omniscient since it can determine the real state completely from the signal it received, while the second expert is nearly ignorant when $\epsilon$ is close to 0.

The second information structure is symmetric with the first information structure, with $(\mu, k_1, k_2, l_1, l_2) = (0.5, 0.5 - \epsilon, 0, 0.5 + \epsilon, 1)$ and posteriors $(b_{1L}, b_{2L}, b_{1H}, b_{2H}) = (0.5 - \epsilon, 0, 0.5 + \epsilon, 1)$. In this information structure, the first expert is nearly ignorant when $\epsilon$ is close to 0, and the second expert is omniscient.

Now consider the distribution that the real information can be the first or the second one with equal probability, which means the more informed expert is chosen to be expert 1 or expert 2 with equal probability. Consider the case when two experts receive different signals. When $(s_1, s_2) = (L, H)$, the agent always observes the input $(a_1, a_2, p_1, p_2) = (0, 1, 0.5 - \epsilon, 0.5 + \epsilon)$ regardless of the real information structure; and when $(s_1, s_2) = (H, L)$, the agent always observes $(a_1, a_2, p_1, p_2) = (1, 0, 0.5 + \epsilon, 0.5 - \epsilon)$. Since the agent does not know who the omniscient expert is, The optimal aggregator in $F_{+2}$ is independent of the specific outputs generated by these inputs. However, the benchmark can always identify the omniscient expert according to the knowledge of the real information structure. Thus, the relative regret of any aggregator in $F_{+2}$ regarding this distribution of information structures is at least

$$2 \times \frac{1}{2} \times \frac{1}{2} \times \left(\frac{1}{2} - \epsilon\right) \times (1 - 0) = \frac{1}{4} - \epsilon.$$

Since $\epsilon$ can be arbitrarily small, any aggregator in $F_{+2}$ cannot guarantee a regret of less than 0.25 regarding this distribution of information structures, which implies the theorem.

## C.5 Proof of Theorem 4.6

Similar to the proof of Theorem 4.4, to compute the maximum loss of $f_{uni}$, we now consider all seven possible different cases of $\pi$.

*Case 1:* $\mu k_1 > (1 - \mu)l_1, \mu k_2 > (1 - \mu)l_2$. In this case, we have $a_i(s) = 1$ for all $i = 1, 2$ and $s = L, H$, and the uniform aggregator always chooses action 1. Meanwhile, we notice in this case that

$$\pi(\omega = 1 \mid S_1 = L, S_2 = H) = \frac{\mu k_1(1 - k_2)}{\mu k_1(1 - k_2) + (1 - \mu)l_1(1 - l_2)} \geq \frac{1}{2},$$

$$\pi(\omega = 1 \mid S_1 = H, S_2 = L) = \frac{\mu(1 - k_1)k_2}{\mu(1 - k_1)k_2 + (1 - \mu)(1 - l_1)l_2} \geq \frac{1}{2},$$

$$\pi(\omega = 1 \mid S_1 = H, S_2 = H) = \frac{\mu(1 - k_1)(1 - k_2)}{\mu(1 - k_1)(1 - k_2) + (1 - \mu)(1 - l_1)(1 - l_2)} \geq \frac{1}{2}.$$

Therefore, the only possible difference between the uniform aggregator and the benchmark is under the condition that $S_1 = S_2 = L$, and the regret is bounded by the following program:

$$\begin{aligned}
\max \quad & -\mu k_1 k_2 + (1 - \mu) l_1 l_2, \\
\text{s.t.} \quad & \mu k_1 k_2 \leq (1 - \mu) l_1 l_2, \\
& \mu k_1 \geq (1 - \mu) l_1, \mu k_2 \geq (1 - \mu) l_2, \\
& 0 \leq k_1 \leq l_1 \leq 1, 0 \leq k_2 \leq l_2 \leq 1, 0 \leq \mu \leq 1.
\end{aligned}$$

And the optimum of the above is $3 - 2\sqrt{2} \approx 0.1716$, which is reached when $\mu = \sqrt{2}/2, k_1 = k_2 = \sqrt{2} - 1, l_1 = l_2 = 1$.

*Case 2:* $\mu(1 - k_1) < (1 - \mu)(1 - l_1), \mu(1 - k_2) < (1 - \mu)(1 - l_2)$. This case is equivalent to Case 1 by substituting $\mu$ with $1 - \mu$ and $k_i$ with $1 - l_i$ for $i = 1, 2$. Thus, they have the same optimum $3 - 2\sqrt{2}$.

*Case 3:* $\mu k_1 \leq (1 - \mu) l_1, \mu(1 - k_1) \geq (1 - \mu)(1 - l_1), \mu(1 - k_2) < (1 - \mu)(1 - l_2)$. In this case, we have $a_1(L) = 0, a_1(H) = 1$, and $a_2(L) = a_2(H) = 0$. We also observe that

$$\pi(\omega = 1 \mid S_1 = L, S_2 = L) = \frac{\mu k_1 k_2}{\mu k_1 k_2 + (1 - \mu) l_1 l_2} \leq \frac{1}{2},$$

$$\pi(\omega = 1 \mid S_1 = L, S_2 = H) = \frac{\mu k_1 (1 - k_2)}{\mu k_1 (1 - k_2) + (1 - \mu) l_1 (1 - l_2)} \leq \frac{1}{2}.$$

Thus, the uniform aggregator matches the benchmark when $S_1 = L$. On the other hand, when $S_1 = H$, there is a split between two experts. Hence, our uniform aggregator would choose action 0.5. We further have

$$\pi(\omega = 1 \mid S_1 = H, S_2 = H) = \frac{\mu(1 - k_1)(1 - k_2)}{\mu(1 - k_1)(1 - k_2) + (1 - \mu)(1 - l_1)(1 - l_2)} \geq \frac{1}{2},$$

As a result, the benchmark will adopt action 1 when $S_1 = H, S_2 = H$. But the action of benchmark when $S_1 = H, S_2 = L$ is unsure. The regret is bounded by the following program:

$$\begin{aligned}
\max \quad & \frac{1}{2} |\mu(1 - k_1) k_2 - (1 - \mu)(1 - l_1) l_2| + \frac{1}{2} (\mu(1 - k_1)(1 - k_2) - (1 - \mu)(1 - l_1)(1 - l_2)), \\
& \mu k_1 \leq (1 - \mu) l_1, \mu(1 - k_1) \geq (1 - \mu)(1 - l_1), \mu(1 - k_2) \leq (1 - \mu)(1 - l_2), \\
& 0 \leq k_1 \leq l_1 \leq 1, 0 \leq k_2 \leq l_2 \leq 1, 0 \leq \mu \leq 1.
\end{aligned}$$

The above program has a maximum value of 0.25 when $\mu = 0.5, k_1 = 0, k_2 = 0.5, l_1 = 1$ and $l_2 = 0.5$.

*Case 4:* $\mu k_1 \leq (1 - \mu) l_1, \mu(1 - k_1) \geq (1 - \mu)(1 - l_1), \mu k_2 > (1 - \mu) l_2$. This case is equivalent to Case 3 by substituting $\mu$ with $1 - \mu$ and $k_i$ with $1 - l_i$ for $i = 1, 2$, and they have the same optimum $1/4$.

*Case 5:* $\mu(1 - k_1) < (1 - \mu)(1 - l_1), \mu k_2 \leq (1 - \mu) l_2, \mu(1 - k_2) \geq (1 - \mu)(1 - l_2)$. This case is equivalent to Case 3 by swapping the order of two experts, and we omit it.

*Case 6:* $\mu k_1 > (1 - \mu) l_1, \mu k_2 \leq (1 - \mu) l_2, \mu(1 - k_2) \geq (1 - \mu)(1 - l_2)$. This case is equivalent to Case 4 by swapping the order of two experts, and we omit it.

*Case 7:* $\mu k_1 \leq (1 - \mu) l_1, \mu(1 - k_1) \geq (1 - \mu)(1 - l_1), \mu k_2 \leq (1 - \mu) l_2, \mu(1 - k_2) \geq (1 - \mu)(1 - l_2)$. We have $a_1(L) = a_2(L) = 0$ and $a_1(H) = a_2(H) = 1$. Meanwhile,

$$\pi(\omega = 1 \mid S_1 = L, S_2 = L) = \frac{\mu k_1 k_2}{\mu k_1 k_2 + (1 - \mu) l_1 l_2} \leq \frac{1}{2},$$

$$\pi(\omega = 1 \mid S_1 = H, S_2 = H) = \frac{\mu(1 - k_1)(1 - k_2)}{\mu(1 - k_1)(1 - k_2) + (1 - \mu)(1 - l_1)(1 - l_2)} \geq \frac{1}{2}.$$

Thus, the uniform aggregator agrees with the benchmark when $S_1 = S_2$. On the other hand, when $S_1 \neq S_2$, there is a split between two experts. Hence, our uniform aggregator will adopt action 0.5. Also, the action of the benchmark is unsure. The following program bounds the regret:

$$\begin{aligned}
\max \quad & \frac{1}{2} |\mu(1 - k_1) k_2 - (1 - \mu)(1 - l_1) l_2| + \frac{1}{2} |\mu k_1 (1 - k_2) - (1 - \mu) l_1 (1 - l_2)|, \\
& \mu k_1 \leq (1 - \mu) l_1, \mu(1 - k_1) \geq (1 - \mu)(1 - l_1) \\
& \mu k_2 \leq (1 - \mu) l_2, \mu(1 - k_2) \geq (1 - \mu)(1 - l_2), \\
& 0 \leq k_1 \leq l_1 \leq 1, 0 \leq k_2 \leq l_2 \leq 1, 0 \leq \mu \leq 1.
\end{aligned}$$

The above program has a maximum value of 0.25 when $\mu = 0.5, k_1 = 0.5, k_2 = 0, l_1 = 0.5, l_2 = 1$.

Synthesizing all 7 cases, we achieve that $L(f_{uni}) \leq 0.25$. Combining with Theorem 4.5, we finish the proof of $L(f_{uni}) = 0.25$.

# D MISSING PROOFS IN SECTION 5

## D.1 Proof of Theorem 5.2

We now give two carefully selected information structures in HOI and demonstrate that no deterministic aggregator in $F_{+2}$ can guarantee a regret less than $3 - 2\sqrt{2}$ regarding both information structures.

| $\mu_1 = 1 - \sqrt{2}/2$ | $\pi_1(s = L \mid \omega = 1)$ | $\pi_1(s = L \mid \omega = 0)$ | $\mu_2 = \sqrt{2}/2$ | $\pi_2(s = L \mid \omega = 1)$ | $\pi_2(s = L \mid \omega = 0)$ |
|---|---|---|---|---|---|
| Experts | 0 | $2 - \sqrt{2}$ | Experts | $3\sqrt{2} - 4$ | $2\sqrt{2} - 2$ |

For the first information structure, we set $(\mu, k, l) = (1 - \sqrt{2}/2, 0, 2 - \sqrt{2})$, which leads to the posterior $(b_L, b_H) = (0, 1/2)$. In this information structure, both experts demonstrate absolute certainty in determining the state when they observe the signal $L$. However, when they observe the signal $H$, neither expert exhibits a preference or inclination towards either state. Therefore, they will recommend action 0 when observing $L$ and action 1 when observing $H$.

For the second information structure, we let $(\mu, k, l) = (\sqrt{2}/2, 3\sqrt{2} - 4, 2\sqrt{2} - 2)$, leading to the posterior $(b_L, b_H) = (\sqrt{2} - 1, \sqrt{2} - 1/2)$. In this information structure, both experts hold partial knowledge about the state. Also, they will recommend action 0 when observing $L$ and action 1 when observing $H$.

Notice that inputs $(1, 0, \sqrt{2}/2, \sqrt{2} - 1)$ and $(0, 1, \sqrt{2} - 1, \sqrt{2}/2)$ appear under both information structures. However, in the first information structure, these inputs only occur when the real state is 1, and each happens with probability $3 - 2\sqrt{2}$. In the second information structure, with probability $17\sqrt{2} - 24$, each input happens with the real state 0. Also, with probability $27 - 19\sqrt{2}$, each input happens with the real state 1. Therefore, if the deterministic aggregator outputs 0 for both inputs, it will cause a regret of $102 - 72\sqrt{2} \approx 0.1766 \geq 3 - 2\sqrt{2} \approx 0.1716$ in the second information structure. If it outputs 1 for both inputs, it will cause a regret of $6 - 4\sqrt{2} \approx 0.3431$ in the first information structure. If it outputs 1 for one of the inputs and 0 for another, it will cause a regret of $3 - 2\sqrt{2}$. Therefore, no deterministic aggregator can guarantee a regret of less than $3 - 2\sqrt{2}$ regarding both information structures above, which implies the theorem.

## D.2 Proof of Theorem 5.4

Since the experts are homogeneous, the follow-the-first-expert is equivalent to the uniform aggregator regarding any information structure in HOI. Therefore, $L_{\text{HOI}}(f_{ftfe}) = 3 - 2\sqrt{2}$ by Theorem 5.6.

## D.3 Proof of Theorem 5.5

We here give a special information structure in HOI, regarding which any aggregator in $\Delta(F_{+2})$ cannot achieve a regret below $3 - 2\sqrt{2}$, which implies the lower bound.

| $\mu_1 = \sqrt{2}/2$ | $\pi_1(s = L \mid \omega = 1)$ | $\pi_1(s = L \mid \omega = 0)$ |
|---|---|---|
| Experts | $\sqrt{2} - 1$ | 1 |

For the information structure, we set $(\mu, k, l) = (\sqrt{2}/2, \sqrt{2} - 1, 1)$, which leads to the posterior $(b_L, b_H) = (1/2, 1)$. In this information structure, two experts always recommend action 1 regardless of the realized signal.

Thus, the agent always observes the input $(1, 1, 1, 1)$ and can only give the same action output regardless of the realized signal. Since the prior is larger than $1/2$, the optimal aggregator in $\Delta(F_{+2})$ should output 1 for this input. However, when $(s_1, s_2) = (L, L)$, the benchmark will adopt the action 0, which leads to the relative regret of any aggregator in $\Delta(F_{+2})$ regarding this information structure at least

$$(1 - \frac{\sqrt{2}}{2}) \times (1 - 0) + \frac{\sqrt{2}}{2} \times (\sqrt{2} - 1)^2 \times (0 - 1) = 3 - 2\sqrt{2},$$

which implies the theorem.

## D.4 Proof of Theorem 5.6

Similar to previous proofs, to compute the maximum regret of $f_{uni}$, we now consider all possible three different cases of $\pi$. Again, we recall that $k_1 \leq l_1$ and $k_2 \leq l_2$ hold by Lemma C.1.

*Case 1:* $\mu k > (1 - \mu)l$. In this case, we have $a_i(s) = 1$ for all $i = 1, 2$ and $s = L, H$. The uniform aggregator always chooses action 1. Also, we can obtain

$$\pi(\omega = 1 \mid S_1 = L, S_2 = H) = \frac{\mu k(1 - k)}{\mu k(1 - k) + (1 - \mu)l(1 - l)} \geq \frac{1}{2},$$

$$\pi(\omega = 1 \mid S_1 = H, S_2 = L) = \frac{\mu(1 - k)k}{\mu(1 - k)k + (1 - \mu)(1 - l)l} \geq \frac{1}{2},$$

$$\pi(\omega = 1 \mid S_1 = H, S_2 = H) = \frac{\mu(1 - k)(1 - k)}{\mu(1 - k)(1 - k) + (1 - \mu)(1 - l)(1 - l)} \geq \frac{1}{2}.$$

Therefore, the only possible difference between the uniform aggregator and the benchmark is under the condition that $S_1 = S_2 = L$, and the regret is bounded by the following program:

$$\max \quad -\mu k^2 + (1 - \mu)l^2,$$

$$\text{s.t.} \quad \mu k^2 \leq (1 - \mu)l^2, \mu k \geq (1 - \mu)l$$

$$0 \leq k \leq l \leq 1, 0 \leq \mu \leq 1.$$

And the optimum of the above is $3 - 2\sqrt{2} \approx 0.1716$, which is reached when $\mu = \sqrt{2}/2, k = \sqrt{2} - 1, l = 1$.

*Case 2:* $\mu(1 - k) < (1 - \mu)(1 - l)$. This case is equivalent to Case 1 by substituting $\mu$ with $1 - \mu$ and $k$ with $1 - l$. Thus they have the same optimum $3 - 2\sqrt{2}$.

*Case 3:* $\mu k \leq (1 - \mu)l, \mu(1 - k) \geq (1 - \mu)(1 - l)$. We have $a_1(L) = a_2(L) = 0$ and $a_1(H) = a_2(H) = 1$. Meanwhile,

$$\pi(\omega = 1 \mid S_1 = L, S_2 = L) = \frac{\mu k^2}{\mu k^2 + (1 - \mu)l^2} \leq \frac{1}{2},$$

$$\pi(\omega = 1 \mid S_1 = H, S_2 = H) = \frac{\mu(1 - k)^2}{\mu(1 - k)^2 + (1 - \mu)(1 - l)^2} \geq \frac{1}{2}.$$

Thus, the uniform aggregator agrees with the benchmark when $S_1 = S_2$. On the other hand, when $S_1 \neq S_2$, there is a split between two experts. Hence, our uniform aggregator will adopt action 0.5. Also, the action of the benchmark is unsure. The following program bounds the regret:

$$\max \quad |\mu(1 - k)k - (1 - \mu)(1 - l)l|,$$

$$\mu k \leq (1 - \mu)l, \mu(1 - k) \geq (1 - \mu)(1 - l)$$

$$0 \leq k \leq l \leq 1, 0 \leq \mu \leq 1.$$

The above program has a maximum value of $3 - 2\sqrt{2}$ when $\mu = \sqrt{2}/2, k = \sqrt{2} - 1, l = 1$.

Synthesizing all three cases, we achieve that $L(f_{uni}) \leq 3 - 2\sqrt{2}$. Combining with Theorem 5.5, we finish the proof of $L(f_{uni}) = 3 - 2\sqrt{2}$.

# E  MISSING PROOFS IN SECTION 6

## E.1  Proof of Theorem 6.5

By Lemma 3.2, to bound the expected regret of any random aggregator in $\Delta(F_{+1})$, it suffices to analyze the expected regret of aggregators in $F_{+1}$ relative to the same benchmark, where information structures are drawn from a carefully selected distribution.

We then give a distribution $D \in \Delta(\text{NHI})$ over two information structures in NHI, regarding which any aggregator in $F_{+1}$ cannot achieve a regret below $3 - 2\sqrt{2} \approx 0.1716$.

| $\mu_1 = \sqrt{2}/2$ | $\pi_1(s = L \mid \omega = 1)$ | $\pi_1(s = L \mid \omega = 0)$ | | $\mu_2 = 1 - \sqrt{2}/2$ | $\pi_2(s = L \mid \omega = 1)$ | $\pi_2(s = L \mid \omega = 0)$ |
|---|---|---|---|---|---|---|
| Experts | $\sqrt{2} - 1 - \epsilon$ | 1 | | Experts | 0 | $2 - \sqrt{2} + \epsilon$ |

For the first information structure, we set $(\mu, k, l) = (\sqrt{2}/2, \sqrt{2} - 1 - \epsilon, 1)$, which leads to the posterior $(b_L, b_H) = ((2 - \sqrt{2} - \sqrt{2}\epsilon)/(4 - 2\sqrt{2} - \sqrt{2}\epsilon), 1)$. $\epsilon$ is a small positive number that is less than $\sqrt{2} - 1$. Experts are sure about the state under this information structure when observing signal $H$. However, when observing signal $L$, they are uncertain about the state when $\epsilon$ is close to 0.

We then construct the second information structure, which is symmetric with the first one. Specifically, we take $(\mu, k, l) = (1 - \sqrt{2}/2, 0, 2 - \sqrt{2} + \epsilon)$ and $(b_L, b_H) = (0, (2 - \sqrt{2})/(4 - 2\sqrt{2} - \sqrt{2}\epsilon))$. Experts completely know the state in this information structure when observing signal $L$; while when observing signal $H$, the two states are indistinguishable when $\epsilon$ is close to 0.

Now consider the distribution that the real information can be the first or the second one with equal probability, which means the more informed signal is $L$ or $H$ with equal probability. Consider the case when two experts receive different signals. When $(s_1, s_2) = (L, H)$, the agent always observes the input $(a_1, a_2) = (0, 1)$ regardless of the real information structure; vice versa for $(s_1, s_2) = (H, L)$. Since the agent

does not know which the more informed signal is, the optimal aggregator in $F_{+1}$ is independent of the specific outputs generated by these inputs. However, the benchmark can always identify the more informed signal according to the knowledge of the real information structure. Thus, the relative regret of any aggregator in $F_{+1}$ against this distribution of information structures is at least

$$\frac{1}{2} \times 2 \times \frac{\sqrt{2}}{2} \times (\sqrt{2} - 1 - \epsilon) \times (2 - \sqrt{2} + \epsilon) \times (1 - 0) = 3 - 2\sqrt{2} - (\frac{3\sqrt{2}}{2} - 2)\epsilon - \frac{\sqrt{2}}{2}\epsilon^2.$$

Since $\epsilon$ can be arbitrarily small, no aggregator in $F_{+1}$ can guarantee a lower regret than $3 - 2\sqrt{2}$ regarding this distribution of information structures. This finishes the proof.

## E.2 Proof of Theorem 6.7

Again, we use Lemma 3.2, and now give a distribution $D \in \Delta(\texttt{NHI})$ over two information structures in NHI, regarding which any aggregator in $F_{+1}$ cannot achieve a regret below $1/6 \approx 0.1667$.

| $\mu_1 = 3/4 - \epsilon$ | $\pi_1(s = L \mid \omega = 1)$ | $\pi_1(s = L \mid \omega = 0)$ |
|---|---|---|
| Experts | $1/3 - 8\epsilon/(9 - 12\epsilon)$ | $1$ |

| $\mu_2 = 1/4 + \epsilon$ | $\pi_2(s = L \mid \omega = 1)$ | $\pi_2(s = L \mid \omega = 0)$ |
|---|---|---|
| Experts | $0$ | $2/3 + 8\epsilon/(9 - 12\epsilon)$ |

For the first information structure, we set $(\mu, k, l) = (3/4 - \epsilon, 1/3 - 8\epsilon/(9 - 12\epsilon), 1)$, which leads to the posterior $(b_L, b_H) = ((3 - 4\epsilon - 8(3\epsilon - 4\epsilon^2)/(3 - 4\epsilon))/(6 + 8\epsilon - 8(3\epsilon - 4\epsilon^2)/(3 - 4\epsilon)), 1)$. $\epsilon < 1/4$. In this information structure, the experts know exactly the state when observing signal $H$. However, when observing signal $L$, it is hard for them to clarify the state when $\epsilon$ is close to 0.

The second information structure is symmetric with the first one. We take $(\mu, k, l) = (1/4 + \epsilon, 0, 2/3 + 8\epsilon/(9 - 12\epsilon))$ and $(b_L, b_H) = (0, (3 + 12\epsilon)/(6 + 8\epsilon - 8(3\epsilon - 4\epsilon^2)/(3 - 4\epsilon)))$. In this information structure, the experts are clear about the state when observing signal $L$. However, when observing signal $H$, they are uncertain about the state when $\epsilon$ is close to 0.

Now consider the distribution that the real information can be the first or the second one with equal probability, i.e., the more informed signal is $L$ or $H$ with equal probability. Consider the case when two experts receive different signals. When $(s_1, s_2) = (L, H)$, the agent always observes the input $(a_1, a_2) = (0, 1, 1/3 - 8\epsilon/(9 - 12\epsilon), 2/3 + 8\epsilon/(9 - 12\epsilon))$ regardless of the real information structure. Instead, when $(s_1, s_2) = (H, L)$, the agent always observes the input $(a_1, a_2) = (1, 0, 2/3 + 8\epsilon/(9 - 12\epsilon), 1/3 - 8\epsilon/(9 - 12\epsilon))$. Similar to the proof of Theorem 6.5, since the agent does not know which the more informed signal is, the optimal aggregator in $F_{+2}$ is independent of the specific outputs generated by these inputs. However, the benchmark can always identify the more informed signal according to the knowledge of the real information structure. Consequently, the relative regret of any aggregator in $F_{+2}$ against this distribution of information structures is at least

$$\frac{1}{2} \times 2 \times (\frac{3}{4} - \epsilon) \times (\frac{1}{3} - \frac{8\epsilon}{9 - 12\epsilon}) \times (\frac{2}{3} + \frac{8\epsilon}{9 - 12\epsilon}) \times (1 - 0) = \frac{1}{6} - \frac{2\epsilon}{9} - \frac{18\epsilon - 32\epsilon^2}{9(3 - 4\epsilon)^2}.$$

Since $\epsilon$ can be arbitrarily small, no aggregator in $F_{+2}$ can guarantee a lower regret than $1/6 \approx 0.1667$ by Lemma 3.2.

## E.3 Proof of Lemma 6.8

Notice that the prediction of the expert who recommends action 0 is

$$\frac{\mu k(1 - k) + (1 - \mu)l(1 - l)}{\mu k + (1 - \mu)l},$$

which is referred to as $p^0$. On the other hand, the prediction of the expert who recommends action 1 is

$$\frac{\mu(1 - k)^2 + (1 - \mu)(1 - l)^2}{\mu(1 - k) + (1 - \mu)(1 - l)},$$

which is referred to as $p^1$. Thus, $p^1 \geq p^0$ naturally holds, and the equality happens when $k = l$.

## E.4 Proof of Proposition 6.9

For proof of (a), by Lemma C.1, $k \leq l$ holds. For every information structure in NHI, we have $a_1(H) = a_2(H) = 1$ and $a_1(L) = a_2(L) = 0$, which implies $\mu k < (1 - \mu)l$ and $\mu(1 - k) \geq (1 - \mu)(1 - l)$. Therefore, $\mu k^2 < (1 - \mu)l^2$ and $\mu(1 - k)^2 \geq (1 - \mu)(1 - l)^2$ holds. When $a_1 = a_2 = 1$, two experts must observe signal $H$, and the benchmark will obtain a posterior higher than $1/2$. In this way, the best strategy for the aggregator is to adopt action 1. The proof is similar when $a_1 = a_2 = 0$.

For proof of (b), we notice that there must exist a random aggregator $f$ that achieves the lowest regret and satisfies (a), and from there, construct another random strategy aggregator $f'$: $f'(a_1, a_2, p_1, p_2) = f(a_2, a_1, p_2, p_1)$ for any input $(a_1, a_2, p_1, p_2)$. Since two experts are

homogeneous, we have

$$
\begin{aligned}
L(f, \pi) &= \sum_{\omega, s_1, s_2} \pi(\omega, s_1, s_2)(u(\phi(\pi(\omega = 1 \mid s_1, s_2)), \omega) - f(a_1(s_1), a_2(s_2), p_1(s_1), p_2(s_2))u(1, \omega) \\
&\quad - (1 - f(a_1(s_1), a_2(s_2), p_1(s_1), p_2(s_2))u(0, \omega)) \\
&= \sum_{\omega, s_1, s_2} \pi(\omega, s_2, s_1)(u(\phi(\pi(\omega = 1 \mid s_2, s_1)), \omega) - f(a_1(s_1), a_2(s_2), p_1(s_1), p_2(s_2))u(1, \omega) \\
&\quad - (1 - f(a_1(s_1), a_2(s_2), p_1(s_1), p_2(s_2))u(0, \omega)) \\
&= \sum_{\omega, s_2, s_1} \pi(\omega, s_2, s_1)(u(\phi(\pi(\omega = 1 \mid s_2, s_1)), \omega) - f'(a_1(s_2), a_2(s_1), p_1(s_2), p_2(s_1))u(1, \omega) \\
&\quad - (1 - f'(a_1(s_2), a_2(s_1), p_1(s_2), p_2(s_1))u(0, \omega)) \\
&= L(f', \pi).
\end{aligned}
$$

Therefore, $f$ and $f'$ achieve the same regret regarding any information structure, which implies $f'$ is also the best random aggregator. Further, notice that the loss function is linear of $f$, thus

$$
L\left(\frac{f + f'}{2}, \pi\right) = \frac{1}{2}L(f, \pi) + \frac{1}{2}L(f', \pi),
$$

which implies $(f + f')/2$ also achieves the lowest regret. Since $(f + f')/2$ satisfies (b), we finish the proof of (b).

For proof of (c), now that there must exist a random aggregator $f$ that achieves the lowest regret and satisfies (a) and (b), we construct another random strategy aggregator $f^\circ$: $f^\circ(a_1, a_2, p_1, p_2) = 1 - f(1 - a_1, 1 - a_2, 1 - p_1, 1 - p_2) = 1 - f(1 - a_2, 1 - a_1, 1 - p_2, 1 - p_1)$ for any input $(a_1, a_2, p_1, p_2)$. Also, for any information structure $\pi$, we can construct information structure $\pi^\circ$ by substituting $\mu, k, l$ with $1 - \mu, 1 - l, 1 - k$. We obtain

$$
\begin{aligned}
L(f, \pi) &= \sum_{\omega, s_1, s_2} \pi(\omega, s_1, s_2)(u(\phi(\pi(\omega = 1 \mid s_1, s_2)), \omega) - f(a_1(s_1), a_2(s_2), p_1(s_1), p_2(s_2))u(1, \omega) \\
&\quad - (1 - f(a_1(s_1), a_2(s_2), p_1(s_1), p_2(s_2))u(0, \omega)) \\
&= \sum_{\omega, s_1, s_2} \pi^\circ(1 - \omega, \bar{s}_1, \bar{s}_2)(u(\phi(\pi(\omega = 1 \mid \bar{s}_1, \bar{s}_2)), 1 - \omega) - f^\circ(a_1(\bar{s}_1), a_2(\bar{s}_2), p_1(\bar{s}_1), p_2(\bar{s}_2))u(1, 1 - \omega) \\
&\quad - (1 - f^\circ(a_1(\bar{s}_1), a_2(\bar{s}_2), p_1(\bar{s}_1), p_2(\bar{s}_2))u(0, 1 - \omega)) \\
&= L(f^\circ, \pi^\circ),
\end{aligned}
$$

where $\bar{s}$ represents the complement signal of $s$. Therefore, $f$ and $f^\circ$ achieve the same regret, which implies $f^\circ$ also attains the lowest regret. Thus,

$$
L\left(\frac{f + f^\circ}{2}, \pi\right) = \frac{1}{2}L(f, \pi) + \frac{1}{2}L(f^\circ, \pi),
$$

which implies $(f + f^\circ)/2$ is also the optimal random aggregator. Since $(f + f^\circ)/2$ satisfies (c), we finish the proof of (c).

At last, (d) holds naturally by (b) and (c).

## E.5 Proof of Theorem 6.10

To compute the maximum regret of $f_{bir}$, we now consider all possible cases of $\pi$ under the conditions. Here, besides the conditions that $k_1 \leq l_1$ and $k_2 \leq l_2$ as given by Lemma C.1, from the definition of NHI, we also know that $\mu k < (1 - \mu)l$ and $\mu(1 - k) \geq (1 - \mu)(1 - l)$.

Further, when $S_1 = S_2 = L$, two experts both recommend action 0, so our aggregator adopts action 0, which is the same as the benchmark. When $S_1 = S_2 = H$, the aggregator and the benchmark both take action 1. Therefore, no regret will be caused when $S_1 = S_2$.

When two experts observe different signals and recommend different actions, it is without loss of generality to assume that $a_1 = 1, a_2 = 0$ due to the symmetry of the aggregator and information structures. Therefore, we have

$$
p_1 = \frac{\mu(1 - k)^2 + (1 - \mu)(1 - l)^2}{\mu(1 - k) + (1 - \mu)(1 - l)}, \quad p_2 = \frac{\mu k(1 - k) + (1 - \mu)l(1 - l)}{\mu k + (1 - \mu)l}.
$$

and $p_1 \geq p_2$ always holds.

*Case 1:* $p_1 + p_2 < 0.98$. The aggregator outputs $\min\{1, (p_1 - 0.6)^2 + (p_2 - 0.4)^2 + 0.5\}$ in this case; however, the benchmark's output is unsure. Therefore, the regret is bounded by the two programs below:

$$
\max \quad ((1 - \mu)l(1 - l) - \mu k(1 - k)) \min\{1, (p_1 - 0.6)^2 + (p_2 - 0.4)^2 + 0.5\},
$$
$$
\text{s.t.} \quad \mu k(1 - k) \le (1 - \mu)l(1 - l), \mu k \le (1 - \mu)l, \mu(1 - k) \ge (1 - \mu)(1 - l)
$$
$$
0 \le k \le l \le 1, 0 \le \mu \le 1,
$$
$$
p_1 + p_2 \le 0.98.
$$

$$
\max \quad (\mu k(1 - k) - (1 - \mu)l(1 - l))(1 - \min\{1, (p_1 - 0.6)^2 + (p_2 - 0.4)^2 + 0.5\}),
$$
$$
\text{s.t.} \quad \mu k^{(}1 - k) \ge (1 - \mu)l(1 - l), \mu k \le (1 - \mu)l, \mu(1 - k) \ge (1 - \mu)(1 - l)
$$
$$
0 \le k \le l \le 1, 0 \le \mu \le 1,
$$
$$
p_1 + p_2 \le 0.98.
$$

The first program achieves the maximized value of $0.08407$ when $\mu = 0.2424, k = 0, l = 0.68$. The second program achieves the maximized value of $0.08402$ when $\mu = 0.7175, k = 0.3938, l = 1$. Therefore, the maximum regret in this case is $0.0841$.

*Case 2:* $p_1 + p_2 > 1.02$. The aggregator outputs $\max\{0, 0.5 - (p_1 - 0.6)^2 - (p_2 - 0.4)^2 + 0.5\}$ in this case, however the benchmark's output is unsure. Therefore, the regret is bounded by the two programs below:

$$
\max \quad ((1 - \mu)l(1 - l) - \mu k(1 - k)) \max\{0, 0.5 - (p_1 - 0.6)^2 - (p_2 - 0.4)^2\},
$$
$$
\text{s.t.} \quad \mu k(1 - k) \le (1 - \mu)l(1 - l), \mu k \le (1 - \mu)l, \mu(1 - k) \ge (1 - \mu)(1 - l)
$$
$$
0 \le k \le l \le 1, 0 \le \mu \le 1,
$$
$$
p_1 + p_2 \ge 1.02.
$$

$$
\max \quad (\mu k(1 - k) - (1 - \mu)l(1 - l))(1 - \max\{0, 0.5 - (p_1 - 0.6)^2 - (p_2 - 0.4)^2\}),
$$
$$
\text{s.t.} \quad \mu k^{(}1 - k) \ge (1 - \mu)l(1 - l), \mu k \le (1 - \mu)l, \mu(1 - k) \ge (1 - \mu)(1 - l)
$$
$$
0 \le k \le l \le 1, 0 \le \mu \le 1,
$$
$$
p_1 + p_2 \ge 1.02.
$$

The first program achieves the maximized value of $0.08402$ when $\mu = 0.2825, k = 0, l = 0.6062$. The second program achieves the maximized value of $0.08078$ when $\mu = 0.7575, k = 0.32, l = 1$. Therefore, the maximum regret in this case is $0.08402$.

*Case 3:* $0.98 \le p_1 + p_2 \le 1.02$. In this case, the aggregator outputs $0.5$; however, the benchmark's output is unsure. Therefore, the regret is bounded by the two programs below:

$$
\max \quad 0.5((1 - \mu)l(1 - l) - \mu k(1 - k)),
$$
$$
\text{s.t.} \quad \mu k(1 - k) \le (1 - \mu)l(1 - l), \mu k \le (1 - \mu)l, \mu(1 - k) \ge (1 - \mu)(1 - l)
$$
$$
0 \le k \le l \le 1, 0 \le \mu \le 1,
$$
$$
0.98 \le p_1 + p_2 \le 1.02.
$$

$$
\max \quad 0.5(\mu k(1 - k) - (1 - \mu)l(1 - l)),
$$
$$
\text{s.t.} \quad \mu k^{(}1 - k) \ge (1 - \mu)l(1 - l), \mu k \le (1 - \mu)l, \mu(1 - k) \ge (1 - \mu)(1 - l)
$$
$$
0 \le k \le l \le 1, 0 \le \mu \le 1,
$$
$$
0.98 \le p_1 + p_2 \le 1.02.
$$

The first program achieves the maximized value of $0.08409$ when $\mu = 0.2574, k = 0, l = 0.6533$. The second program achieves the maximized value of $0.08409$ when $\mu = 0.7426, k = 0.3467, l = 1$. Therefore, the maximum regret in this case is $0.08409$.

Synthesizing all three cases, we obtain that $L_{\text{NHI}}(f_{bir}) \le 2 \times 0.0841 = 0.1682$. Also, when $\mu = 0.7426, k = 0.3467, l = 1$, the regret of the aggregator regarding this information structure is $0.1682$, which implies the result.

## E.6 Proof of Theorem 6.11

To compute the maximum regret of $f_{alg}$, we now consider all possible cases of $\pi$, under the conditions as pointed out by the proof of Theorem 6.10. Further, it suffices to consider the scenario with $a_1 = 1, a_2 = 0$.

We now divide all possible predictions into 100 intervals: $\{(p_1, p_2) : lb_1 <= p_1 <= lb_1 + 0.1, lb_2 <= p_2 <= lb_2\}$ for all $lb_1, lb_2 \in \{0, 0.1, 0.2, \cdots, 0.9\}$. For each interval, the aggregator is linear, with

$$f(1, 0, lb_1 + \delta_1, lb_2 + \delta_2) = \delta_1 \delta_2 f(1, 0, lb_1 + 0.1, lb_2 + 0.1) + \delta_1(1 - \delta_2) f(1, 0, lb_1 + 0.1, lb_2)$$
$$+ (1 - \delta_1)\delta_2 f(1, 0, lb_1, lb_2 + 0.1) + (1 - \delta_1)(1 - \delta_2) f(0, 1, lb_1, lb_2).$$

Here, $f(1, 0, lb_1 + 0.1, lb_2 + 0.1), f(1, 0, lb_1 + 0.1, lb_2), f(1, 0, lb_1, lb_2 + 0.1)$ and $f(1, 0, lb_1, lb_2)$ are obtained by the algorithm in advance. Therefore, the maximum regret in this interval can be bounded by two programs

$$\max \quad f(1, 0, p_1, p_2)((1 - \mu)l(1 - l) - \mu k(1 - k)),$$
$$\text{s.t.} \quad \mu k(1 - k) \le (1 - \mu)l(1 - l), \mu k \le (1 - \mu)l, \mu(1 - k) \ge (1 - \mu)(1 - l)$$
$$0 \le k \le l \le 1, 0 \le \mu \le 1,$$
$$lb_1 \le p_1 \le lb_1 + 0.1, lb_2 \le p_2 \le lb_2 + 0.1.$$

$$\max \quad (1 - f(1, 0, p_1, p_2))(\mu k(1 - k) - (1 - \mu)l(1 - l)),$$
$$\text{s.t.} \quad \mu k^{(}1 - k) \ge (1 - \mu)l(1 - l), \mu k \le (1 - \mu)l, \mu(1 - k) \ge (1 - \mu)(1 - l)$$
$$0 \le k \le l \le 1, 0 \le \mu \le 1,$$
$$lb_1 \le p_1 \le lb_1 + 0.1, lb_2 \le p_2 \le lb_2 + 0.1.$$

By symmetry, the regret of the aggregator can then be bounded by two times the maximum value of all the programs, which is 0.1673 when $\mu = 0.763, k = 0.3106, l = 1$, given by Wolfram Mathematica.

# F    MISSING PROOFS IN APPENDIX A

## F.1    Proof of Proposition A.1

Since $\phi^t$ is only decided by $t, a_1, a_2, p_1, p_2$ should be the same for every $s_1, s_2$ when considering $\pi$, which implies $L(f^r, \pi, u) = \Delta u_0 \cdot L^t(f^r, \pi)$.

## F.2    Proof of Theorem A.2

We here give a special information structure in HOI, regarding which any aggregator in $\Delta(F_{+2})$ cannot achieve a regret below $(\sqrt{1 + 1/t} - \sqrt{1/t})^2$, which implies the lower bound.

| $\mu_1 = \sqrt{1/t+1}$ | $\pi_1(s = L \mid \omega = 1)$ | $\pi_1(s = L \mid \omega = 0)$ |
|---|---|---|
| Experts | $(\sqrt{t + 1} - 1)/t$ | 1 |

For the information structure, we set $(\mu, k, l) = (\sqrt{1/t+1}, \frac{\sqrt{t+1}-1}{t}, 1)$, which leads to the posterior $(b_L, b_H) = (1/(t+1), 1)$. In this information structure, two experts always recommend action 1 regardless of the realized signal.

Therefore, the agent always observes the input $(1, 1, 1, 1)$ and can only give the same action output regardless of the realized signal. Since the prior is larger than $1/(t + 1)$, the optimal aggregator in $\Delta(F_{+2})$ should output 1 for this input. However, when $(s_1, s_2) = (L, L)$, the benchmark will adopt the action 0, which leads to the relative regret of any aggregator in $\Delta(F_{+2})$ regarding this information structure at least

$$(1 - \frac{1}{\sqrt{t + 1}}) \times (1 - 0) + \frac{1}{\sqrt{t + 1}} \times \left(\frac{\sqrt{t + 1} - 1}{t}\right)^2 \times (0 - t) = \left(\sqrt{1 + \frac{1}{t}} - \sqrt{\frac{1}{t}}\right)^2.$$

This implies the theorem.

## F.3    Proof of Theorem A.3

Similarly, we give a special information structure in HOI, regarding which any aggregator in $\Delta(F_{+2})$ cannot achieve a regret below $(\sqrt{t + t^2} - t)^2$, which implies the lower bound.

| $\mu_1 = 1 - \sqrt{t/t+1}$ | $\pi_1(s = L \mid \omega = 1)$ | $\pi_1(s = L \mid \omega = 0)$ |
|---|---|---|
| experts | 0 | $1 - \sqrt{t}(\sqrt{t + 1} - \sqrt{t}) - \epsilon$ |

For the information structure, we set

$$(\mu, k, l) = \left(1 - \sqrt{t/t+1}, 0, 1 - \frac{t(\sqrt{t + 1} - \sqrt{t})}{\sqrt{t}} - \epsilon\right),$$

which leads to the posterior

$$(b_L, b_H) = \left(0, \frac{\sqrt{t + 1} - \sqrt{t}}{(t + 1)(\sqrt{t + 1} - \sqrt{t}) + \sqrt{t}\epsilon}\right).$$

$\epsilon$ here is a small positive number less than $1 - \sqrt{t}(\sqrt{t+1} - \sqrt{t})$. In this information structure, two experts always recommend action 0 regardless of the realized signal.

Therefore, the agent always observes the input $(0, 0, 0, 0)$ and can only give the same action output regardless of the realized signal. Since the prior is smaller than $1/(t+1)$, the optimal aggregator in $\Delta(F_{+2})$ should output 0 for this input. However, when $(s_1, s_2) = (H, H)$, the benchmark will adopt the action 1, which leads to the relative regret of any aggregator in $\Delta(F_{+2})$ regarding this information structure at least

$$(1 - \frac{t}{\sqrt{t+1}}) \times (t - 0) + \frac{t}{\sqrt{t+1}} \times \left( \frac{t(\sqrt{t+1} - \sqrt{t})}{\sqrt{t}} + \epsilon \right)^2 \times (0 - 1) = (\sqrt{t + t^2} - t)^2 - 2t\sqrt{t+1}(\sqrt{t^2 + t} - 1)\epsilon - \frac{t}{\sqrt{t+1}}\epsilon^2.$$

Since $\epsilon$ can be arbitrarily small, no aggregator in $\Delta(F_{+2})$ can guarantee a lower regret than $(\sqrt{t + t^2} - t)^2$, which implies the theorem.

## F.4 Proof of Theorem A.4

To compute the maximum regret of $f_p$, we now consider all possible three different cases of $\pi$, under the conditions $k_1 \le l_1$ and $k_2 \le l_2$ given by Lemma C.1.

*Case 1: $t\mu k > (1 - \mu)l$.* In this case, we have $a_i(s) = 1$ for all $i = 1, 2$ and $s = L, H$. The aggregator always chooses action 1. Also, we can obtain

$$\pi(\omega = 1 \mid S_1 = L, S_2 = H) = \frac{\mu k(1 - k)}{\mu k(1 - k) + (1 - \mu)l(1 - l)} \ge \frac{1}{t + 1},$$

$$\pi(\omega = 1 \mid S_1 = H, S_2 = L) = \frac{\mu(1 - k)k}{\mu(1 - k)k + (1 - \mu)(1 - l)l} \ge \frac{1}{t + 1},$$

$$\pi(\omega = 1 \mid S_1 = H, S_2 = H) = \frac{\mu(1 - k)(1 - k)}{\mu(1 - k)(1 - k) + (1 - \mu)(1 - l)(1 - l)} \ge \frac{1}{t + 1}.$$

Therefore, the only possible difference between the aggregator and the benchmark is under the condition that $S_1 = S_2 = L$, and the regret is bounded by the following program:

$$\max \quad -t\mu k^2 + (1 - \mu)l^2,$$
$$\text{s.t.} \quad t\mu k^2 \le (1 - \mu)l^2, t\mu k \ge (1 - \mu)l$$
$$0 \le k \le l \le 1, 0 \le \mu \le 1.$$

We can bound the maximum value of the program as follows:

$$-t\mu k^2 + (1 - \mu)l^2 \le l^2 \left( 1 - \mu - \frac{(1 - \mu)^2}{t\mu} \right)$$
$$\le 1 - \mu - \frac{(1 - \mu)^2}{t\mu}$$
$$= 1 + \frac{2}{t} - (1 + \frac{1}{t})\mu - \frac{1}{t\mu}$$
$$\le 1 + \frac{2}{t} - 2\sqrt{\frac{1}{t}(\frac{1}{t} + 1)},$$

which takes equality at $\mu = \sqrt{1/t+1}, k = \frac{\sqrt{t+1} - 1}{t}, l = 1$. Since the above value satisfies the constraints, the optimum of the above program is $(\sqrt{1 + 1/t} - \sqrt{1/t})^2$.

*Case 2: $t\mu(1 - k) < (1 - \mu)(1 - l)$.* Similarly, the regret is bounded by the following program:

$$\max \quad t\mu(1 - k)^2 - (1 - \mu)(1 - l)^2,$$
$$\text{s.t.} \quad t\mu(1 - k)^2 \ge (1 - \mu)(1 - l)^2, t\mu(1 - k) \le (1 - \mu)(1 - l)$$
$$0 \le k \le l \le 1, 0 \le \mu \le 1.$$

We bound the maximum value of the program similarly as the above and obtain that the maximum value is $(\sqrt{t + t^2} - t)^2$ when $\mu = 1 - \sqrt{t/1+t}, k = 0, l = 1 - \frac{t(\sqrt{t+1} - \sqrt{t})}{\sqrt{t}}$.

*Case 3:* $t\mu k \leq (1-\mu)l, t\mu(1-k) \geq (1-\mu)(1-l)$. We have $a_1(L) = a_2(L) = 0$ and $a_1(H) = a_2(H) = 1$. Meanwhile,

$$\pi(\omega = 1 \mid S_1 = L, S_2 = L) = \frac{\mu k^2}{\mu k^2 + (1-\mu)l^2} \leq \frac{1}{t+1},$$

$$\pi(\omega = 1 \mid S_1 = H, S_2 = H) = \frac{\mu(1-k)^2}{\mu(1-k)^2 + (1-\mu)(1-l)^2} \geq \frac{1}{t+1}.$$

Thus, the aggregator agrees with the benchmark when $S_1 = S_2$. On the other hand, when $S_1 \neq S_2$, there is a split between two experts. Hence, our prob-$p$ aggregator will adopt action $p$. Also, the action of the benchmark is unsure. The following two programs bound the regret:

$$\begin{aligned}
\max \quad & 2\left(t\mu(1-k)k - (1-\mu)(1-l)l\right)(1-f), \\
\text{s.t.} \quad & t\mu k(1-k) \geq (1-\mu)l(1-l), \\
& t\mu k \leq (1-\mu)l, t\mu(1-k) \geq (1-\mu)(1-l), \\
& 0 \leq k \leq l \leq 1, 0 \leq \mu \leq 1.
\end{aligned}$$

and

$$\begin{aligned}
\max \quad & 2\left(-t\mu(1-k)k + (1-\mu)(1-l)l\right)f, \\
\text{s.t.} \quad & t\mu k(1-k) \leq (1-\mu)l(1-l), \\
& t\mu k \leq (1-\mu)l, t\mu(1-k) \geq (1-\mu)(1-l), \\
& 0 \leq k \leq l \leq 1, 0 \leq \mu \leq 1.
\end{aligned}$$

For the first program, we bound the maximum value as follows:

$$\begin{aligned}
2(t\mu k(1-k) - (1-\mu)l(1-l))(1-p) &\leq 2t\mu k(1-k)(1-p) \\
&\leq 2tk(1-k)\frac{1}{tk+1}(1-p) \\
&= 2(1-p)\left(-k + \left(1 + \frac{1}{t}\right) - \frac{1 + \frac{1}{t}}{tk+1}\right) \\
&\leq 2(1-p)\left(\sqrt{1 + \frac{1}{t}} - \sqrt{\frac{1}{t}}\right)^2,
\end{aligned}$$

which takes equality at $\mu = \sqrt{1/t+1}, k = \frac{\sqrt{1+t}-1}{t}, l = 1$. Since the above value satisfies the constraints, the optimum of the above program is $2(1-p)\left(\sqrt{1 + 1/t} - \sqrt{1/t}\right)^2$.

Similarly, the maximum value of the second program is $2p(\sqrt{t+t^2} - t)^2$ when $\mu = 1 - \sqrt{t/1+t}, k = 0, l = 1 - \frac{t(\sqrt{t+1}-\sqrt{t})}{\sqrt{t}}$.

We also have that $\left(\sqrt{1 + 1/t} - \sqrt{1/t}\right)^2 \geq (\sqrt{t+t^2} - t)^2$ for any $t \geq 1$ and $\left(\sqrt{1 + 1/t} - \sqrt{1/t}\right)^2 \leq (\sqrt{t+t^2} - t)^2$ for any $t \leq 1$. Synthesizing all three cases, we achieve that $L(f_p) \leq \left(\sqrt{1 + 1/t} - \sqrt{1/t}\right)^2$ for any $p \in \left[0.5, \frac{\left(\sqrt{1+1/t}-\sqrt{1/t}\right)^2}{2(\sqrt{t+t^2}-t)^2}\right]$. Combining with Theorem A.2, we finish the proof.

## F.5 Proof of Theorem A.5

Similar to the proof in Theorem A.4, the problem can be divided into three cases, and the solution is the same as above. Synthesizing all three cases, we achieve that $L_{\text{HOI}}^t(f_p) \leq (\sqrt{t+t^2} - t)^2$ for any $p \in \left[1 - \frac{(\sqrt{t+t^2}-t)^2}{2\left(\sqrt{1+1/t}-\sqrt{1/t}\right)^2}, 0.5\right]$. Combining with Theorem A.3, we finish the proof.

## F.6 Proof of Theorem A.6

By Lemma 3.2, we now give a distribution $D \in \Delta(\text{NHI})$ over two information structures in NHI, regarding which any aggregator in $F_{+1}$ cannot achieve a regret below $\frac{2\left(\sqrt{1+1/t}-\sqrt{1/t}\right)^2(\sqrt{t+t^2}-t)^2}{(\sqrt{t+t^2}-t)^2+\left(\sqrt{1+1/t}-\sqrt{1/t}\right)^2}$.

| $\mu_1 = \sqrt{1/t+1}$ | $\pi_1(s = L \mid \omega = 1)$ | $\pi_1(s = L \mid \omega = 0)$ |
|---|---|---|
| Experts | $(\sqrt{t+1} - 1)/t - \epsilon$ | $1$ |

| $\mu_2 = 1 - \sqrt{t/t+1}$ | $\pi_2(s = L \mid \omega = 1)$ | $\pi_2(s = L \mid \omega = 0)$ |
|---|---|---|
| Experts | 0 | $1 - \sqrt{t}(\sqrt{t+1} - \sqrt{t})$ |

For the first information structure, we set $(\mu, k, l) = (\sqrt{1/t+1}, \frac{\sqrt{t+1}-1}{t} - \epsilon, 1)$, which leads to the posterior $(b_L, b_H) = (\frac{\sqrt{t+1}-1-t\epsilon}{(t+1)(\sqrt{t+1}-1)-t\epsilon}, 1)$. $\epsilon$ can be any small positive number that is less than $\frac{\sqrt{t+1}-1}{t}$. Experts are sure about the state in this information structure when observing signal $H$. However, when observing signal $L$, it is nearly uncertain which action is better.

We then construct the second information structure symmetric with the first one: $(\mu, k, l) = (1 - \sqrt{t/t+1}, 0, 1 - \frac{t(\sqrt{t+1}-\sqrt{t})}{\sqrt{t}})$ and $(b_L, b_H) = (0, 1/t+1)$. Similarly, here, experts know exactly the state when observing signal $L$. Nevertheless, when observing signal $H$, it is nearly uncertain which action is better.

Now consider the distribution that the real information is the first one with the probability of

$$\frac{(\sqrt{t+t^2} - t)^2}{(\sqrt{t+t^2} - t)^2 + \left(\sqrt{1+\frac{1}{t}} - \sqrt{\frac{1}{t}}\right)^2 - \epsilon^2 - (1 - \frac{2(\sqrt{t+1}-1)}{t})\epsilon}$$

and the second one with the probability of

$$\frac{\left(\sqrt{1+\frac{1}{t}} - \sqrt{\frac{1}{t}}\right)^2 - \epsilon^2 - (1 - \frac{2(\sqrt{t+1}-1)}{t})\epsilon}{(\sqrt{t+t^2} - t)^2 + \left(\sqrt{1+\frac{1}{t}} - \sqrt{\frac{1}{t}}\right)^2 - \epsilon^2 - (1 - \frac{2(\sqrt{t+1}-1)}{t})\epsilon}.$$

Consider the case when two experts receive different signals. When $(s_1, s_2) = (L, H)$, the agent always observes the input $(a_1, a_2) = (0, 1)$ regardless of the real information structure. When $(s_1, s_2) = H, L$, the agent always observes the input $(a_1, a_2) = (1, 0)$ regardless of the real information structure. Since the agent doesn't know which the more informed signal is, the optimal aggregator in $F_{+1}$ is independent of the specific outputs generated by these inputs. However, the benchmark can always identify the more informed signal according to the knowledge of the real information structure. Thus, the relative regret of any aggregator in $F_{+1}$ against this distribution of information structures is at least

$$\frac{\left(\sqrt{1+\frac{1}{t}} - \sqrt{\frac{1}{t}}\right)^2 - \epsilon^2 - (1 - \frac{2(\sqrt{t+1}-1)}{t})\epsilon}{(\sqrt{t+t^2} - t)^2 + \left(\sqrt{1+\frac{1}{t}} - \sqrt{\frac{1}{t}}\right)^2 - \epsilon^2 - (1 - \frac{2(\sqrt{t+1}-1)}{t})\epsilon}$$

$$\times 2 \times \sqrt{\frac{t}{t+1}} \times (1 - \frac{t(\sqrt{t+1} - \sqrt{t})}{\sqrt{t}}) \times \frac{t(\sqrt{t+1} - \sqrt{t})}{\sqrt{t}} \times (1 - 0)$$

Since $\epsilon$ can be arbitrarily small, no aggregator in $F_{+1}$ can guarantee a lower regret than

$$\frac{2\left(\sqrt{1+\frac{1}{t}} - \sqrt{\frac{1}{t}}\right)^2 (\sqrt{t+t^2} - t)^2}{(\sqrt{t+t^2} - t)^2 + \left(\sqrt{1+\frac{1}{t}} - \sqrt{\frac{1}{t}}\right)^2}$$

regarding this distribution of information structures. This implies the theorem.

## F.7 Proof of Theorem A.7

According to Case 3 in proof in Theorem A.4, we achieve that

$$L_{\mathsf{NHI}}^t(f_p) \leq \frac{2\left(\sqrt{1+\frac{1}{t}} - \sqrt{\frac{1}{t}}\right)^2 (\sqrt{t+t^2} - t)^2}{(\sqrt{t+t^2} - t)^2 + \left(\sqrt{1+\frac{1}{t}} - \sqrt{\frac{1}{t}}\right)^2}$$

for

$$p = \frac{\left(\sqrt{1+\frac{1}{t}} - \sqrt{\frac{1}{t}}\right)^2}{(\sqrt{t+t^2} - t)^2 + \left(\sqrt{1+\frac{1}{t}} - \sqrt{\frac{1}{t}}\right)^2}.$$

Combining with Theorem A.6, we finish the proof.

