# OpenReview forum: "Robust Decision Aggregation with Second-order Information"
_ACM.org/TheWebConf/2024/Conference — TheWebConf24_

### Official Review · Reviewer_V5sm · 2023-11-23

**Novelty:** 4
**Technical Quality:** 4

**Review:**

The paper explores a decision aggregation problem with two experts providing binary recommendations based on private signals about an unknown world state. The authors investigate the impact of incorporating second-order information (each expert's forecast on the other's recommendation) on aggregation performance. Using a minimax regret framework, the analysis reveals that, in general information structures, second-order information provides no benefit. However, positive outcomes are observed when assuming conditionally independent signals. For deterministic aggregators facing heterogeneous experts, a robust aggregator leveraging second-order information outperforms counterparts without it. In cases of homogeneous experts, random aggregators using second-order information can surpass optimal ones under non-degenerate signal assumptions.

Overall, I found the paper to be well-written, and if you take the model as given, the paper gives a fairly satisfying and complete first investigation – the questions asked are exactly the ones I would hope are answered first.


One major modeling concern I have is the assumption that each expert will honestly report the second-order information, especially when two of them are competitors in general. I am a bit confused since the expert’s utility is not clearly discussed in the paper. Could you elaborate a bit about the expert’s behavior model and its validity?

A minor observation is that the paper appears to use "agent" and "aggregator" interchangeably. While it seems that both terms refer to the same entity, I suggest maintaining consistency in the terminology or explicitly indicating the interchangeability of these terms at some point in the paper.

**Questions:**

Can you provide insights into the challenges or difficulties associated with generalizing your results to scenarios involving more than two experts, states, and signals?

**Reviewer Confidence:**

2: The reviewer is willing to defend the evaluation, but it is likely that the reviewer did not understand parts of the paper

**Scope:**

3: The work is somewhat relevant to the Web and to the track, and is of narrow interest to a sub-community

---

### Official Review · Reviewer_yFRU · 2023-11-23

**Novelty:** 6
**Technical Quality:** 6

**Review:**

**Summary of the paper**

The paper discusses a decision aggregation problem where two experts each make a binary recommendation of actions based on a private signal about an unknown binary world state. An aggregator, not knowing the joint distribution of the private signals and actual world states (such a joint distribution is known as the information structure), determines which action to choose by aggregating the experts' decisions. This paper particularly discusses whether the second-order information (each expert’s forecast on the other expert’s decision) benefits the aggregator.

The performance of the aggregator is measured by a minimax regret framework and is compared to an omniscient benchmark. Here the regret is defined as the worst-case difference between the omniscient benchmark’s utility and the aggregator’s expected utility.
The authors analyze optimal deterministic and random aggregators both with and without second-order information with various underlying assumptions.

First the authors show that in general information structures, the second-order information has no benefit to the aggregator’s performance. The follow-the-first-expert aggregator, which always agree with the first expert regardless of other information, is proved to be optimal with a regret of 0.5.

Then the authors consider the conditionally independent information structures where experts' signals are independently influenced by the world state.

For deterministic aggregators lacking second-order information, the follow-the-first-expert aggregator achieves the lowest regret of 0.5. In contrast, the threshold aggregator, suited for those with second-order information, aligns with experts' consensus or opts for the 'surprisingly popular' choice in disagreements, attaining a minimum regret of 1/3. For random aggregators, the uniform aggregator minimizes regret to 0.25 by adhering to experts' consensus or equally randomly selecting a decision in case of disagreement.

For homogeneous experts with identical marginal signal distribution, the minimal regret for both deterministic and random aggregators are 3 - 2√2 ≈ 0.1716, which can be achieved by the follow-the-first-expert aggregator and the uniform aggregator. The authors further consider the situation of homogeneous experts with non-degenerate signals, where the experts will recommend different actions when observing different signals. In this context, the second-order information is helpful for random aggregators.

The authors lastly extend the setting to general utility functions.

**Evaluation**

I believe this paper provides valuable contributions to the information aggregation literature. I think the setting with a small group of heterogeneous experts is relevant to many practical applications. Thus, I think the model studied in this paper is interesting. In addition, I am excited about many of the observations made by the authors. I especially like the observations about the usefulness of second-order information in this paper.

Perhaps a limitation of the paper is the restricted numbers of experts and states. The authors also mentioned about the setting with more experts and richer signal structures as a future research direction. Nevertheless, I think the current results in the paper have enough merit for the paper’s publication.

**Minor Issue**

Typo: the expression “the most closed setting” should be “The closest setting” in line 93

**Questions:**

No question.

**Ethics Review Description:**

N.A.

**Reviewer Confidence:**

3: The reviewer is confident but not certain that the evaluation is correct

**Scope:**

3: The work is somewhat relevant to the Web and to the track, and is of narrow interest to a sub-community

---

### Official Review · Reviewer_CVSM · 2023-11-24

**Novelty:** 5
**Technical Quality:** 6

**Review:**

# Summary

This paper considers the robust two-expert information aggregation problem: there are two experts, each of which receives a (possibly correlated) signal from an unknown binary state. Based on their posterior beliefs, each expert recommends a binary action to the principal whose utility is to match the state. The goal is to design an aggregator, which maps experts' reports to the principal's action. Besides aggregators that only rely on experts' recommended actions, the authors also consider aggregators with second-order information (i.e., each expert also reports her posterior belief about the other expert's recommended action). The authors study this problem in a robust framework, i.e., the information structure of experts' signals is unknown to the principal, and the goal is to design the minimax optimal aggregator that minimizes the regret over all possible information structures.

The authors consider various scenarios (general/conditionally independent/homogeneous/homogeneous&non-degenerate information structures) and identify the minimax optimal deterministic/randomized aggregators with/without the second-order information. By comparing their regret guarantees, the authors illustrate whether the second-order information is beneficial in this robust framework.

Besides the matching-state utility, the authors also study an extension for general utility (with binary action, binary state).

# Strengths

This paper introduces a clean economic model. The study is comprehensive and technically interesting. Though some results are straightforward, I like the clear takeaway message of this paper. The overall presentation is good and I enjoy reading this paper.

# Weaknesses

I don’t see significant weakness in the paper itself. That said, the topic and the model (i.e., binary experts, binary state, binary action) might seem stylized and restrictive for some WebConf audiences. I also have some other comments discussed below.


# Comments

In the current model, the first-order information is encoded as the recommended action $a_i$; while the second-order information is a distribution $p_i$ over the other expert's recommended action. Is it without loss of generality to assume that each expert reports a recommended action instead of her posterior belief of the state? Suppose we consider an alternative model where the first-order information is the expert's posterior belief of the state, and the second-order information is the expert's posterior belief of the other expert's posterior belief. Can any results or techniques be extended?

It would be helpful to clarify that the statement "the second-order information is not beneficial" is with respect to the *worst-case* regret. It is possible that there exists an aggregator in $F_{+2}$ with weakly better regret than the robust-optimal aggregators in $F_{+1}$ for *all* information structures, and is strictly better for some information structures.

Do you have any idea whether the results and techniques can be extended beyond two experts?

Though all classes of information structures (i.e., ALL, CI, HOI, NHI) are theoretically natural to study, it would be beneficial if the authors could use real-world examples to motivate each class.

It would be helpful to give some proof sketch for Theorem 3.1 and highlight that the constructed information structure is not conditionally independent, which better motivates the following sections. Similarly, some proof sketch for Theorem 4.5 is also helpful.

**Questions:**

I asked some questions/comments above but please don't feel obligated to respond.

**Reviewer Confidence:**

3: The reviewer is confident but not certain that the evaluation is correct

**Scope:**

3: The work is somewhat relevant to the Web and to the track, and is of narrow interest to a sub-community

---

### Official Review · Reviewer_Ed9U · 2023-11-25

**Novelty:** 5
**Technical Quality:** 5

**Review:**

The paper studies the problem of decision aggregation with two experts who provide binary recommendations and second-order predictions about each other’s recommendations. The authors adopt a minimax regret framework to evaluate the performance of different deterministic/random aggregators and characterize the conditions under which second-order information is beneficial or redundant. The paper also extends the results to the setting with general utility functions.

The paper extends an interesting question of ``how second-order information can enhance decision aggregation in a robust way'' to the setting with only two experts. The authors have provided a comprehensive analysis of different scenarios, such as general, conditionally independent, homogeneous, and non-degenerate information structures, and deterministic and random aggregators. The threshold aggregator introduced by this work is particularly interesting. The paper is well-written, clear, and rigorous. The authors provide intuitive explanations and examples for the main results and techniques.
On the other hand, the paper assumes that the experts are truthful and have access to the joint information structure. It would be interesting to relax these assumptions and study how strategic behavior or incomplete information would affect the aggregation problem.
Moreover, the paper focuses on the binary action and state setting which is a very limited framework for real-world applications. It would be interesting to extend the results to the multi-action and multi-state setting and compare the performance of different aggregators in terms of social welfare or other metrics. Finally, the paper provides no empirical or experimental evaluation of the proposed aggregators. It would be helpful to test the aggregators on real-world data or scenarios and demonstrate their practical implications and limitations.

**Questions:**

- Is it possible to analyze this problem in a setting where the experts only have limited knowledge about the joint information structure? What are the challenges for doing so?
- In the paper, you have assumed that the experts are incentive-compatible. However, you haven't specified the reward structure that incentivizes the experts to be truthful. Can you elaborate on this and explain how incentive compatibility is ensured?
- Have you considered the setting where the predictions and actions are performative, i.e., the actions that the agent/aggregator takes would change the state of the world? To what extent do the techniques and results hold for the performative setting?
- What are some interesting choices for the general utility function in Section 7? Can you give a few examples that would actually make sense for real-world applications?
- Have you considered the possibility that the experts might collude with each other? Are the aggregators robust against such behaviors?

**Reviewer Confidence:**

2: The reviewer is willing to defend the evaluation, but it is likely that the reviewer did not understand parts of the paper

**Scope:**

3: The work is somewhat relevant to the Web and to the track, and is of narrow interest to a sub-community

---

### Official Review · Reviewer_yjeJ · 2023-11-29

**Novelty:** 4
**Technical Quality:** 5

**Review:**

The authors investigate the utility of second-order information within smaller expert groups. Building upon the foundational work of Prelec et al. (2017), which elucidates the application of the surprisingly popular principle and second-order information for deducing true states through crowd wisdom, the study addresses a constraint inherent in the requirement of a sufficiently large expert group for optimal information aggregation. To overcome this limitation, the authors delve into the realm of robust decision aggregation, specifically focusing on scenarios involving only two experts. Employing the minmax regret framework, the authors conduct a comparative analysis of the aggregator's performance against omniscient benchmarks possessing knowledge of the joint information structure. The findings reveal that the aggregator's performance is contingent upon the information structure. In cases of a general information structure, second-order information confers no discernible benefit, and no random aggregator can guarantee a regret below 0.5. However, within conditionally independent information structures, second-order information emerges as an enhancement, improving the performance of both deterministic and random aggregators, provided certain assumptions hold, including the homogeneity of experts with non-degenerate signals.

Strengths:

1. The obtained results seamlessly complement the antecedent research conducted by Prelec et al. (2017), providing a nuanced and comprehensive analysis of the utility of second-order information within more confined expert groups. The findings contribute to the existing body of knowledge by offering a detailed discussion on the significance and implications of second-order information in scenarios involving smaller expert cohorts.

2. Similarly, the authors provide detailed proof throughout the paper to help readers understand the mechanisms more easily.

Weaknesses:

1. The efficacy of the surprisingly popular principle stems from the convergence of decisions made by an infinite, homogeneous pool of experts to the actual distribution. Consequently, there is some confusion regarding the authors' specific interest in elucidating the value of second-order information within the context of only two experts.

2. Even though this paper provides detailed proof, some theorems lack great intuitive explanations to make them more reasonable.

3. It is not clear how to think of extensions to three or more experts. The value and significance of the results is questionable if all the conclusions and insights are limited to aggregation of two experts.

I have read the rebuttals and updated my review accordingly.

**Questions:**

Questions for Author Response

1. In the Problem Statement Section, the authors assume that the prior of the world state is known to both experts. If the prior for two experts is different, do the main results of this paper change?

2. Similarly, the authors assume that the aggregator lacks precise knowledge of the entire information structure. However, a pertinent question arises: if aggregators possess partial knowledge of the information structure, how would this impact the key findings presented in this paper?

3. Can you provide intuitively accessible explanations elucidating the efficacy of second-order information in the context of information aggregation involving only two experts? (the surprising popular principle does not seem relevant anymore.)

4. Can you comment on generations? What if we have three or more experts? Do the results hold for general state or action spaces?

**Ethics Review Description:**

No significance ethical issues comes to mind.

**Reviewer Confidence:**

3: The reviewer is confident but not certain that the evaluation is correct

**Scope:**

3: The work is somewhat relevant to the Web and to the track, and is of narrow interest to a sub-community

---

### Decision · Program_Chairs · 2024-01-22

**Decision:**

Accept

**Comment:**

This paper considers a robust two-expert information aggregation problem: there are two experts, each of which receives a (possibly correlated) signal from an unknown binary state. Based on their posterior beliefs, each expert recommends a binary action to the principal whose utility is to match the state. The goal is to design an aggregator, which maps experts' reports to the principal's action. The authors also consider aggregators with second-order information (i.e., each expert also reports her posterior belief about the other expert's recommended action). The authors study this problem in a robust framework, i.e., the information structure of experts' signals is unknown to the principal, and the goal is to design the minimax optimal aggregator that minimizes the regret over all possible information structures.

 strength: This paper introduces a clean economic model. The study is comprehensive and technically interesting. Though some results are straightforward, reviewers agree that there is a clear takeaway from the paper. The overall presentation is good.

 weakness: the topic and the model (i.e., binary experts, binary state, binary action) might seem stylized and restrictive for some WebConf audiences.

 The authors did a good job in responding to reviewers comments. With rebuttal from the authors, most of the review concerns are addressed, modulo the following one: the value of second-order information within the context of only two experts remain somewhat unjustified. Reviewers have unanimously agreed that the scope of the work is relevant but narrow. Overall, this is a worthy contribution and could be presented as a poster.